# Retrieval-Augmented Multiple Instance Learning

**Yufei Cui[1✉] , Ziquan Liu[2], Yixin Chen[3], Yuchen Lu[4], Xinyue Yu[4],**
**Xue Liu[1], Tei-Wei Kuo[56], Miguel R. D. Rodrigues[2], Chun Jason Xue[3], Antoni B. Chan[3]**
[1]Mila, McGill University    [2]University College London    [3]City University of Hong Kong
[4] Mila, Université de Montréal [5]National Taiwan University    [6]MBZUAI

## Abstract

Multiple Instance Learning (MIL) is a crucial weakly supervised learning method applied across various domains, e.g., medical diagnosis based on whole slide images (WSIs). Recent advancements in MIL algorithms have yielded exceptional performance when the training and test data originate from the same domain, such as WSIs obtained from the same hospital. However, this paper reveals a performance deterioration of MIL models when tested on an out-of-domain test set, exemplified by WSIs sourced from a novel hospital. To address this challenge, this paper introduces the Retrieval-AugMented MIL (RAM-MIL) framework, which integrates Optimal Transport (OT) as the distance metric for nearest neighbor retrieval. The development of RAM-MIL is driven by two key insights. First, a theoretical discovery indicates that reducing the input's intrinsic dimension can minimize the approximation error in attention-based MIL. Second, previous studies highlight a link between input intrinsic dimension and the feature merging process with the retrieved data. Empirical evaluations conducted on WSI classification demonstrate that the proposed RAM-MIL framework achieves state-of-the-art performance in both in-domain scenarios, where the training and retrieval data are in the same domain, and more crucially, in out-of-domain scenarios, where the (unlabeled) retrieval data originates from a different domain. Furthermore, the use of the transportation matrix derived from OT renders the retrieval results interpretable at the instance level, in contrast to the vanilla $l_2$ distance, and allows for visualization for human experts.

## 1 Introduction

As the standard supervised learning paradigm, single instance learning has been the focus of machine learning research and its performance has been improved significantly since the advent of deep learning [1]. Nonetheless, a notable drawback of single instance learning lies in its reliance on a substantial volume of labeled data [2], which poses practical challenges as a result of the high expense and time-consuming nature of fine-grained data labeling. Consequently, multiple instance learning (MIL) [3] has gained increasing popularity, given its ability to be trained with limited supervision. This trend has particularly gained momentum within the domain of histopathology, where the analysis of medical images at a gigabyte scale, such as whole slide images (WSI), has emerged as a focal point in recent studies [4, 5, 6, 7, 8, 9, 10, 11].

Even though the performance of MIL on cancer diagnosis [13, 12] is remarkable, existing algorithms are only evaluated on the in-domain test set and have the risk of performance degradation when confronted with distribution shifts between the training and test sets [14, 15]. The distribution shift problem is particularly important in the context of automated medical diagnosis models, since the models are deployed across diverse hospitals in distinct regions, each may employing varying imaging

---

[1]E-mail: yufeicui92@gmail.com, yufei.cui@mila.quebec
Code can be found at `https://github.com/ralphc1212/ram-mil`.

37th Conference on Neural Information Processing Systems (NeurIPS 2023).

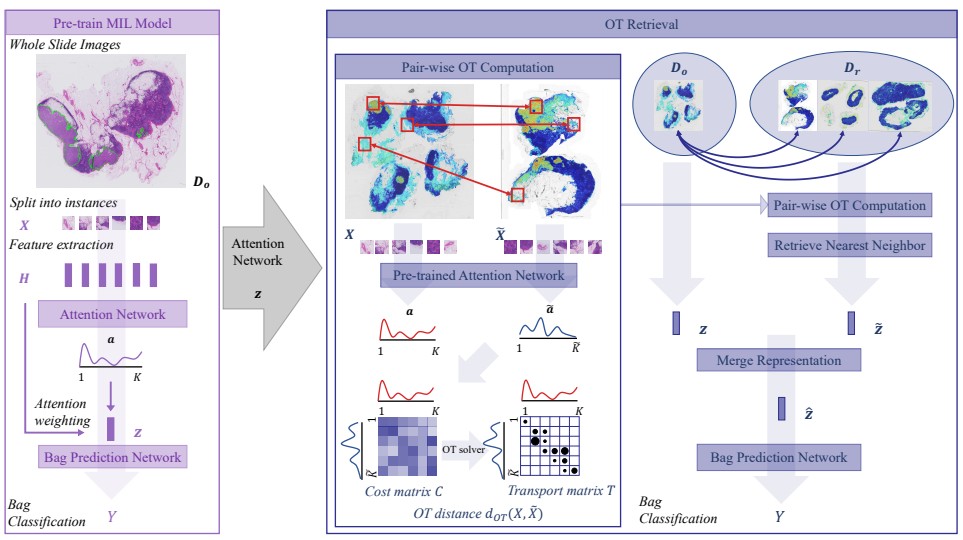

Figure 1: The proposed RAM-MIL method. **Left**: An attention-based MIL model is pre-trained on the training set to produce feature representations and attention weights. **Right**: With pre-trained features and attention weights, Optimal-Transport (OT) is employed to compute bag-to-bag distance, based on which the nearest neighbor bag from the retrieval set is selected. A training bag's feature is then merged with its retrieved bag, which is used as the input feature to a bag classifier.

techniques. The robustness of such models in the face of distribution shift is of utmost importance within this safety-critical application, as any failure to address this challenge may result in severe consequences.

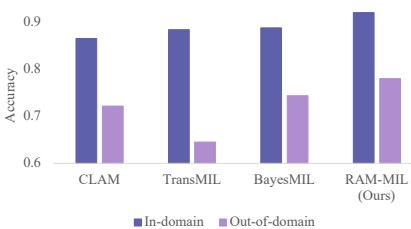

Figure 2: The comparisons of accuracy of MIL methods under in-domain and out-of-domain (OOD) settings. The out-of-domain is tested on Camelyon17 [12] with models trained on Camelyon16 [13]. MIL fails to generalize well to OOD data, while the proposed RAM-MIL improves the OOD performance as well as the in-domain performance.

This paper first unveils that the performance degradation indeed happens for state-of-the-art MIL models (Fig. 2) and proposes a retrieval-based MIL solution for improving both in-domain and out-of-domain performance. We prove a theorem that demonstrates the reduced input dimensions leads to improved MIL performance. Inspired by the connection between feature merging learning and intrinsic dimension [16], we propose the Retrieval-AugMented MIL (RAM-MIL) algorithm to learn a low intrinsic dimension feature space and enhance the model generalization, especially for out-of-domain data. In the retrieval process, we use the optimal transport (OT) as the distance metric between two bags because our main theorem is proved with the generalized OT measure. In our setting, the instances in a bag and their associated attention values in attention-based MIL are used to form a discrete distribution used to represent a bag, and OT is used to calculate the distance between the two discrete distributions of two bags. Following retrieval, the original bags are merged with their retrieved counterparts from a retrieval dataset, employing a merge function such as convex combination, to generate feature representations. Subsequently, a classifier is trained as the MIL model. The retrieval dataset can either be the original dataset in the absence of out-of-domain data, or an *unlabeled* out-of-domain dataset in the out-of-domain adaptation setting. Our empirical study confirms that RAM-MIL effectively reduces the intrinsic dimensionality of the input feature space, surpassing existing MIL methods in both in-domain and out-of-domain settings. The advantage of OT distance over $l_2$ and OT-variant distances is demonstrated by our ablation study. In addition, the OT-based retrieval result can be visualized using the transportation matrix, which makes the retrieval process interpretable. Our contributions are three-fold:

1. This work is among the first to investigate the out-of-domain performance for MIL, which is a vital issue for the application of MIL in automated medical diagnosis with WSIs. Our empirical study exposes the risk of recent MIL algorithms under distribution shifts.
2. Our theoretical result based on Wasserstein distance in the input space shows a negative correlation between MIL performance and the input data dimension, which motivates us

to propose the novel RAM-MIL framework based on the OT as the distance metric in the retrieval process.

3. Our experiment demonstrates that RAM-MIL outperforms state-of-the-art MIL methods in both in-domain and out-of-domain adaptation scenarios. Additionally, the transportation matrix of OT distance provides a tool to visualize the correspondence between original and retrieval instances, which makes our retrieval method interpretable.

## 2 Motivation of RAM-MIL

In this section, we show that why using OT could lead to a performance improvement for multiple instance learning.

We follow the standard formulation of Attention-based Multiple Instance Learning (MIL) [3, 4]. In MIL, the input is a bag of instances, $\boldsymbol{X} = \{\boldsymbol{x}_1, \ldots, \boldsymbol{x}_K\}, \boldsymbol{x}_k \in \mathbb{R}^D, \boldsymbol{X} \in \mathcal{X}$ and $K$ is the number of instances, which varies for different bags. There is a bag-level label $Y$, which is positive if at least one of the instances is positive, and negative if all instances are negative. We further assume the instances also have corresponding instance-level labels $\{y_1, \ldots, y_K\}$, which are *unknown* during training. There are $N$ such bag-label pairs constituting the dataset $\mathcal{D} = \{\boldsymbol{X}_n, Y_n\}_{n=1}^N$. The objective of MIL is to learn an optimal function for predicting the bag-level label with the bag of instances as input. To this end, the MIL model should be able to aggregate the information of instances $\{\boldsymbol{x}_k\}_{k=1}^K$ to make the final decision. A well-adopted aggregation method is the embedding-based approach which maps $\boldsymbol{X}$ to a bag-level representation $\boldsymbol{z} \in \mathbb{R}^d$ and use $\boldsymbol{z}$ to predict $Y$. [3] extends the embedding-based aggregation approach by leveraging the attention mechanism, namely attention-based deep MIL (ABMIL). First, a transformation $g(\cdot)$ computes a low-dimensional embedding $\boldsymbol{h}_k = g(\boldsymbol{x}_k) \in \mathbb{R}^d$ for each instance $\boldsymbol{x}_k$. The attention module aggregates the set of embeddings $\{\boldsymbol{h}_k\}_{k=1}^K$ into a bag level embedding $\boldsymbol{z} = \sum_{k=1}^K a_k \boldsymbol{h}_k = \sum_{k=1}^K a_k g(\boldsymbol{x}_k) = \mathcal{A}(\boldsymbol{X})$, where the attention $a_k$ for the $k$-th instance is computed via a softmax function,

$$a_k = \frac{\exp\{\boldsymbol{m}^T(\tanh(\boldsymbol{V}_1^T\boldsymbol{h}_k)\odot\mathrm{sigmoid}(\boldsymbol{V}_2^T\boldsymbol{h}_k))\}}{\sum_{j=1}^K \exp\{\boldsymbol{m}^T(\tanh(\boldsymbol{V}_1^T\boldsymbol{h}_j)\odot\mathrm{sigmoid}(\boldsymbol{V}_2^T\boldsymbol{h}_j))\}}. \tag{1}$$

where $\boldsymbol{m}, \boldsymbol{V}_1, \boldsymbol{V}_2$ are the network parameters. The bag embedding $\boldsymbol{z}$ is then mapped to the logits $\boldsymbol{u}$ with a feed forward layer with parameter $\boldsymbol{W}$ for the bag-level classification, $\boldsymbol{u} = \boldsymbol{W}^T\boldsymbol{z}$.

Assuming there is a ground-truth set scoring function $S(\boldsymbol{X}) : \mathcal{X} \mapsto \mathbb{R}$ that generates the label $Y$, our task is to approximate the function with the attention-based model. We have the following theorem demonstrating that, when employing the Wasserstein distance in the input space, the approximation error of a Lipschitz continuous set scoring function is upper bounded by a term associated with the input dimension $d$. The insight from the theorem offers a guidepost for the design of our methodology.

**Theorem 1.** *Suppose $S(\cdot)$ is a Lipschitz continuous function with respect to Wasserstein distance $\mathcal{W}_p$ ($1 \leq p < \infty$) and the Lipschitz constant is $L$. The bag of instances $\boldsymbol{X}$ is sampled from a probability distribution $\mu(\boldsymbol{x})$ with distribution dimension $d_\mu$ (intrinsic dimension). For any invertible map $\Phi : \mathcal{X} \to \mathbb{R}^d$, $\exists$ function $\sigma$ and $\gamma$, such that for any $\boldsymbol{X} \in \mathcal{X}$,*

$$|S(\boldsymbol{X}) - \gamma(\Phi_{\boldsymbol{X}\in\mathcal{X}}\{\sigma(\boldsymbol{x}) : \boldsymbol{x} \in \boldsymbol{X}\})| \leq O(L \cdot K^{-\frac{1}{d_\mu}}). \tag{2}$$

The theorem has two implications for the ABMIL problem. First, using the Wasserstein metric in the input space, we establish this dimension-dependent error bound for the approximation, indicating the good property of Wasserstein metric in the theoretical sense. Second, this dimension-dependent error bound motivates our RAM-MIL framework to reduce the intrinsic dimensionality of the input space for improving the performance of ABMIL, which is consistent with a previous work [16] that shows feature mixup is effective at reducing input feature dimension. This paper uses OT distance as the instantiation of the Wasserstein distance, since OT with Euclidean distance as the cost is equivalent to $\mathcal{W}_1$. We use the entropy-regularized OT as the approximation of OT distance in our implementation since the regularization makes the optimization process more stable and efficient and is widely used in machine learning and vision research [17, 18]. The incorporation of OT into Multiple Instance Learning (MIL) provides a new framework for understanding and assessing the similarity between different instances. Intuitively, each bag in MIL can be considered as a probability distribution of instances, where the probabilities are the attention values $\boldsymbol{a}$. Thus, OT offers a rigorous and meaningful metric to quantify the similarity between two distributions. It takes into account not only the individual differences of instances but also their global arrangement within a bag, hence,

**Algorithm 1** Retrieval-AugMented Multiple Instance Learning (RAM-MIL) Algorithm

1: Pre-train an MIL model on $\mathcal{D}_o = \{\boldsymbol{X}_n, Y_n\}_{n=1}^{N_o}$.
2: Extract the sets of instance representations, i.e., $\mathcal{H}_o = \{\boldsymbol{H}_n\}_{n=1}^{N_o}$ and $\mathcal{H}_r = \{\tilde{\boldsymbol{H}}_m\}_{m=1}^{N_r}$, and compute their attentions, i.e., $\{\boldsymbol{a}_n\}_{n=1}^{N_o}$ and $\{\tilde{\boldsymbol{a}}_m\}_{m=1}^{N_r}$
3: Extract the sets of bag representations $\{\boldsymbol{z}_n\}_{n=1}^{N_o}$ and $\{\tilde{z}_n\}_{n=1}^{N_r}$ with pre-trained model.
4: **Input:** $\{\boldsymbol{X}_n, Y_n, \boldsymbol{H}_n, \boldsymbol{z}_n, \boldsymbol{a}_n\}_{n=1}^{N_o}, \{\tilde{\boldsymbol{X}}_m, \tilde{Y}_m, \tilde{\boldsymbol{H}}_m, \tilde{z}_m, \tilde{\boldsymbol{a}}_m\}_{m=1}^{N_r}$
5: **for** $n$ from 1 to $N_o$ **do**
6:     $\nu^* = \arg\min_{\tilde{\boldsymbol{H}}_m \in \mathcal{H}_r} d_{OT}(\mu_n, \nu_m)$, where $d_{OT}(\mu_n, \nu_m)$ is solved by (4)
7:     Store $\hat{z}_n = \pi(\boldsymbol{z}_n, \boldsymbol{z}_{m^*})$, where $m^* = \text{getIndex}(\nu^*)$
8: **end for**
9: Train a logistic regression bag classifier on $\{\hat{z}_n, Y_n\}_{n=1}^{N_r}$.

providing a holistic comparison. Note that although Theorem 1 is proved for the input space $\mathcal{X}$, it is trivial to prove a similar result for the feature space $\mathcal{H}$ (see the supplemental).

Building upon the benefits brought by OT, we propose a retrieval mechanisms to further strengthen the MIL framework. The key intuition is to leverage the rich information available across different bags. This allows for more robust and generalizable learning as each bag might not exist in isolation but is part of a broader set of data with shared and contrasting characteristics. The retrieval process, particularly when guided by OT, helps to identify and bring into context "neighbor" bags that are most relevant or similar to a given bag. This allows the model to learn more effectively by taking into account the broader context in which each bag exists. Empirically, we confirm the OT-guided retrieval could effectively reduce the intrinsic dimension of input feature space, which explains the performance of proposed methodology. Moreover, with the computed transport matrix, the correspondence between the original and selected bag can be visualized, which makes the retrievial and classification process more interpretable/explainable.

## 3 Methodology

This section presents the RAM-MIL algorithm based on OT distance and its application in the out-of-domain adaptation.

### 3.1 Retrieval-Augmented Multiple Instance Learning based on Optimal Transport

We introduce the methodology that utilizes OT distance to address the retrieval problem for multiple instance learning (MIL). The idea of enhancing MIL using retrieval lies in augmenting a bag feature from the original dataset with its nearest neighbor bag feature from the retrieval dataset. Consider the original dataset of bags marked for model training and validation, denoted as $\mathcal{D}_o = \{\boldsymbol{X}_n, Y_n\}_{n=1}^{N_o}$, and the unlabeled retrieval set represented as $\mathcal{D}_r = \{\tilde{\boldsymbol{X}}_m\}_{m=1}^{N_r}$, where $N_o$ and $N_r$ signify the number of bags in the two respective sets. For each instance $\boldsymbol{x}_k$ in a bag $\boldsymbol{X}$, we extract an instance representation $\boldsymbol{h}_k = g(\boldsymbol{x}_k)$, where $g(\cdot)$ is a neural net for feature extraction, such as ResNet50 [19]. Recall that an MIL model maps the bag of instance representations to the logits $\boldsymbol{u}$ for loss computation, expressed as $\boldsymbol{u} = \boldsymbol{W}^T \boldsymbol{z}$. The intermediate representation $\boldsymbol{z} = \mathcal{A}(\boldsymbol{X})$, computed by averaging instance features and attention weights $\boldsymbol{z} = \mathcal{A}(\boldsymbol{X}) = \sum_{k=1}^{K} a_k \boldsymbol{h}_k = \sum_{k=1}^{K} a_k g(\boldsymbol{x}_k)$, is regarded as the *bag representation*.

The objective of the retrieval scheme is to select the bag in the retrieval set that is nearest to the input bag. To perform bag-to-bag retrieval, we use the distributional distance between two bags, which considers their constituent instances. Once the retrieved bag is obtained, its bag representation $\tilde{\boldsymbol{z}}^*$ is then merged with the original bag representation $\boldsymbol{z}$. The merged feature, denoted as $\hat{\boldsymbol{z}} = \pi(\boldsymbol{z}, \tilde{\boldsymbol{z}}^*)$, is used for bag classification , where $\pi(\cdot, \cdot)$ is the merging function, e.g., a convex combination.

**Retrieval with Optimal Transport.** To leverage the probabilistic geometry between instances, instead of directly retrieving the nearest bags based on the bag representations $\boldsymbol{z}$, our retrieval operates on the bags of instance representations $\mathcal{H}_r = \{\tilde{\boldsymbol{H}}_m\}_{m=1}^{N_r} = \{\{\tilde{\boldsymbol{h}}_{mk}\}_{k=1}^{K}\}_{m=1}^{N_r}$. Here $K$ represents the number of instances, $\tilde{\boldsymbol{H}}_m$ indicates *the bag of instance representations* and $\tilde{\boldsymbol{h}}_{mk}$ is an *instance representation* derived via $\tilde{\boldsymbol{h}}_{mk} = g(\tilde{\boldsymbol{x}}_{mk})$. For each bag in $\mathcal{D}_o$, we similarly extract bags of instance representations as $\mathcal{H}_o$.

For each bag $\boldsymbol{H} \in \mathcal{H}_o$, we find its nearest neighbor in $\mathcal{H}_r$ using the following OT problem to calculate distance between the two bags $\boldsymbol{H}$ and $\tilde{\boldsymbol{H}} \in \mathcal{H}_r$. Let $\mu = \sum_{i=1}^{K} a_i \delta_{\boldsymbol{h}_i}$ and $\nu = \sum_{j=1}^{\tilde{K}} \tilde{a}_j \delta_{\tilde{\boldsymbol{h}}_j}$ be

discrete distributions supported on $\boldsymbol{H} = \{\boldsymbol{h}_k\}_{k=1}^{K}$ and $\tilde{\boldsymbol{H}} = \{\tilde{\boldsymbol{h}}_k\}_{k=1}^{\tilde{K}}$, where $\delta_{\boldsymbol{\mu}} = \delta(\boldsymbol{h} - \boldsymbol{\mu})$ is the translated Dirac delta function. Here, $a_i$ and $\tilde{a}_j$ are the attention scores from the MIL model for instance $i$ in bag $\boldsymbol{H}$ and instance $j$ in bag $\tilde{\boldsymbol{H}}$, respectively, forming the probability vectors for the bags. The OT problem between these two distributions can be formulated as follows:

$$d_{\mathrm{OT}}(\mu, \nu) = \min_{\boldsymbol{T} \in \mathcal{T}(\boldsymbol{a}, \tilde{\boldsymbol{a}})} \sum_{i=1}^{K} \sum_{j=1}^{\tilde{K}} c(\boldsymbol{h}_i, \tilde{\boldsymbol{h}}_j) T_{ij} \tag{3}$$

In this equation, $\boldsymbol{T}$ denotes the transport plan matrix where each element $T_{ij}$ specifies the amount of "mass" to be transported from $\boldsymbol{h}_i$ to $\tilde{\boldsymbol{h}}_j$. The function $c(\boldsymbol{h}_i, \tilde{\boldsymbol{h}}_j)$ is a cost function that quantifies the cost of transporting a unit of mass from $\boldsymbol{h}_i$ to $\tilde{\boldsymbol{h}}_j$. A common choice of $c(\boldsymbol{h}_i, \tilde{\boldsymbol{h}}_j)$ is the squared $l_2$ distance between the instance features, i.e., $c(\boldsymbol{h}_i, \tilde{\boldsymbol{h}}_j) = \|\boldsymbol{h}_i - \tilde{\boldsymbol{h}}_j\|_2^2$. $\mathcal{T}(\boldsymbol{a}, \tilde{\boldsymbol{a}})$ represents the set of all matrices $\boldsymbol{T}$ that satisfy the marginals $\boldsymbol{T}^\top \mathbf{1}_K = \boldsymbol{a}$ and $\boldsymbol{T}\mathbf{1}_{\tilde{K}} = \tilde{\boldsymbol{a}}$. Here, $\mathbf{1}_n$ and $\mathbf{1}_m$ are vectors of ones. We also impose an entropy regularization [20] term to reduce the sensitivity to outlier instances. Specifically, we solve the following OT problem with a Sinkhorn's algorithm [20]:

$$d_{\mathrm{OT}}(\mu, \nu) = \min_{\boldsymbol{T} \in \mathcal{T}(\boldsymbol{a}, \tilde{\boldsymbol{a}})} \sum_{i=1}^{K} \sum_{j=1}^{\tilde{K}} c(\boldsymbol{h}_i, \tilde{\boldsymbol{h}}_j) T_{ij} + \beta \cdot \sum_{ij} T_{ij} \log T_{ij} \tag{4}$$

$$s.t., \boldsymbol{T}^\top \mathbf{1}_K = \boldsymbol{a}, \boldsymbol{T}\mathbf{1}_{\tilde{K}} = \tilde{\boldsymbol{a}}, \boldsymbol{T} \geq 0$$

In the context of the retrieval-based MIL problem, the nearest neighbor bag for a given bag $\boldsymbol{H}$ would be the one $\tilde{\boldsymbol{H}}$ in the retrieval set that results in the smallest OT cost with $\boldsymbol{H}$:

$$\nu^* = \underset{\tilde{\boldsymbol{H}}_m \in \mathcal{H}_r}{\arg\min} \, d_{OT}(\mu, \nu_m) \tag{5}$$

In the retrieval process, we use an MIL model pre-trained on $\mathcal{D}_o$ to extract the attention vectors $\boldsymbol{a}$ and $\tilde{\boldsymbol{a}}$, which serve as mass probability distributions for calculating the OT, in addition to extracting bag representations $\boldsymbol{z}$ and $\tilde{\boldsymbol{z}}$. This method effectively transforms the instance importance, as determined by the model, into a mass probability distribution. Instances with high attention scores are considered to possess larger "mass". In the context of the OT problem, such instances become more "expensive" to transport, thereby significantly influencing the computation of the transport cost. Consequently, the model is encouraged to pair similar instances carrying substantial mass, a strategy that aligns with the intuitive notion of matching similar instances together. Employing the pre-extracted attention to classify the weighted bag representation after retrieval ensures consistency with the retrieval process and enhances efficiency. The whole retrieval-enhanced multiple instance learning is shown in Algo. 1.

### 3.2 Unsupervised Domain Adaptation

The proposed methodology can be effectively employed in the context of unsupervised domain adaptation (UDA) [21]. UDA seeks to leverage labeled data from a source domain to train a model that can perform well on an unlabeled target domain.

Remarkably, our retrieval algorithm (as described in Algo. 1) can be seamlessly applied to the UDA task. To achieve this, we set $\mathcal{H}_o$ to represent the source domain and $\mathcal{H}_r$ to represent the target domain. Our approach retrieves the nearest neighbor bag representation from the target domain for each bag representation in the source domain. Subsequently, we train the final classifier using the merged representation and the source domain label.

The RAM-MIL based on OT allows leveraging the geometric structure of the instance space across domains. This promotes the alignment of the source and target domains at the instance level. By merging the bag features, the discriminative information embedded in the source domain is transferred to the target domain. As a result, the bag classifier is able to discriminate the target domain bags without needing access to their labels.

### 3.3 Reduction of Intrinsic Dimension

In this section, we validate our hypothesis that retrieval can effectively reduce the intrinsic dimension of each bag's distribution. We use the two NN estimator [22] for computing the intrinsic dimension for the bag representations, either the original features for MIL or merged features for retrieval models. Figure 3 shows the intrinsic dimension of CLAM [4], Manifold Mixup [16], OT based

retrieval and different variants of retrieval methods. The $l_2$ distance based retrieval, approximate OT retrieval and Hausdorff distance based retrieval are variants of our retrieval method that will be elaborated in Section 5.

The figure shows that our proposed retrieval methods indeed lower the intrinsic dimensions of bag representations in both in-domain and out-of-domain settings. As shown in (2), the approximation performance of a ground-truth set scoring function is theoretically constrained by the intrinsic dimension of the distribution, when measured under the Wasserstein distance. RAM-MIL delivers the lowest intrinsic dimensions among the methods compared, thus validating its superior performance in classification and aligns with our theoretical analysis (2). We offer more detailed discussion of intrinsic dimension in Section 5.2.

Figure 3: The intrinsic dimension of bag representations for different MIL methods and retrieval models. CLAM uses the original feature and the rest use merged feature from retrieval. RAM-MIL is the most effective method at reducing intrinsic dimension of feature space.

## 4   Related works

**Attention-based multiple instance learning**
ABMIL [3] first introduced the attention mechanism for MIL to an interpretation of importance for instances. Building on this, DSMIL [9] used contrastive learning for feature extraction and established global connections between instance attentions. TransMIL [5] took this a step further and developed a correlated MIL, employing multi-head self-attention and spatial information encoding for thorough global correlation. CLAM [4] extends ABMIL to the case of multiple classes and builds an integrated toolbox for visualizing the uncertainty. DTFD-MIL [23] proposes a two-stage feature distillation MIL framework for enhancing the performance. Bayes-MIL [11] studies the fundamental interpretability problem in the attention based MIL framework and proposed to address it with a probabilistic method. In our RAM-MIL, the attention weight is utilized as a measure of the probability density, indicating the amount of "mass" being moved from the source bag to the target bag. Greater attention weights in RAM-MIL indicate more significant relationships between instances in the bags being compared.

**Domain adaptation for medical imaging** There have been a large volume of research studying the domain adaptation in medical imaging [24, 25, 26]. However, most of the preceding studies on whole slide images domain generalization have primarily focused on instance-level classification [27, 28, 29, 30, 31, 32, 33, 34, 35, 36, 37, 38, 39], rather than tackling the more complex challenge of weakly-supervised bag-level classification (MIL). In contrast, our approach seeks to address this more complex problem of unsupervised domain adaptation for MIL. We review these methods in details and make extensive comparisons in the supplemental.

**Retrieval method for whole slide images** Recent works also study how to efficiently retrieve relative WSI from database. Yottixel [40] builds a search engine for indexing WSIs at scale. SISH [41] uses a tree structure for fast search of WSI followed by an uncertainty-based ranking algorithm for retrieval. In experiment, we integrate the open-sourced SISH as a search engine into our RAM-MIL pipeline for comparisons (See supplemental). HHOT [42] proposes to use optimal transport (OT) as a distance measure to compare different WSIs, or different WSI datasets. Our paper, focusing on classification tasks, studies the principle why OT is suitable for WSI classification and propose a retrieval-based classification process. We compare RAM-MIL and HHOT in our experiments.

## 5   Experiments

The proposed methodology is evaluated on whole slide image (WSI) datasets (Camelyon16 [43, 13], Camelyon17 [12], TCGA-NSCLC, CPTAC-UCEC and CPTAC-LSCC) and general MIL datasets (See results in supplemental). The evaluation is divided into in-domain and UDA settings. For the in-domain setting, the retrieval set $\mathcal{H}_r$ is the union of training and validation set of $\mathcal{H}_o$. For the UDA setting, the retrieval set is a held-out dataset with no labels involved in training or inference.

Table 1: Results on CAMELYON16 and CAMELYON17 for in-domain classification and unsupervised domain adaptation, for comparisons with related methods.

| | In-Domain (CAM16) | | Out-of-Domain (CAM17) | |
| | AUC | Accuracy | AUC | Accuracy |
|---|---|---|---|---|
| ABMIL | $0.9010\pm0.026$ | $0.8750\pm0.020$ | $0.7287\pm0.035$ | $0.7190\pm0.049$ |
| DSMIL | $0.8944\pm0.051$ | $0.8682\pm0.060$ | - | - |
| CLAM | $0.9177\pm0.044$ | $0.8650\pm0.060$ | $0.7613\pm0.054$ | $0.7214\pm0.044$ |
| TransMIL | $0.9307\pm0.024$ | $0.8837\pm0.041$ | $0.5697\pm0.118$ | $0.6451\pm0.100$ |
| Bayes-MIL | $0.9432\pm0.049$ | $0.8875\pm0.052$ | $0.7839\pm0.044$ | $0.7435\pm0.058$ |
| HHOT-kNN (k=1) | $0.7007\pm0.035$ | $0.7318\pm0.051$ | $0.6939\pm0.076$ | $0.7173\pm0.061$ |
| HHOT-kNN (k=3) | $0.7618\pm0.026$ | $0.7544\pm0.055$ | $0.7263\pm0.048$ | $0.7523\pm0.076$ |
| Mixup Retr$_\text{I}$ | $0.9260\pm0.051$ | $0.8825\pm0.051$ | - | - |
| Mixup Retr$_\text{IO}$ | $0.9271\pm0.048$ | $0.8850\pm0.046$ | $0.7658\pm0.052$ | $0.7594\pm0.044$ |
| Mixup Retr$_\text{O}$ | $0.9271\pm0.045$ | $0.8825\pm0.048$ | $0.7641\pm0.056$ | $0.7593\pm0.058$ |
| $l_2$ Retr$_\text{I}$ | $0.9281\pm0.047$ | $0.8800\pm0.055$ | - | - |
| $l_2$ Retr$_\text{IO}$ | $0.9241\pm0.048$ | $0.8950\pm0.048$ | $0.7627\pm0.055$ | $0.7353\pm0.051$ |
| $l_2$ Retr$_\text{O}$ | $0.9398\pm0.045$ | $0.8950\pm0.055$ | $0.7738\pm0.050$ | $0.7493\pm0.050$ |
| RAM-MIL Retr$_\text{I}$ | $\mathbf{0.9451\pm0.036}$ | $0.8925\pm0.050$ | - | - |
| RAM-MIL Retr$_\text{IO}$ | $0.9365\pm0.052$ | $\mathbf{0.9200\pm0.050}$ | $\mathbf{0.7974\pm0.054}$ | $0.7433\pm0.073$ |
| RAM-MIL Retr$_\text{O}$ | $0.9419\pm0.048$ | $0.9175\pm0.051$ | $0.7681\pm0.058$ | $\mathbf{0.7795\pm0.021}$ |

We compare our OT-based retrieval with Euclidean distance retrieval, Hausdorff distance retrieval and Manifold MixUp as baselines. The Euclidean ($l_2$ norm) distance $d_{l_2}(\cdot,\cdot)$ and Hausdorff distance based retrieval are implemented in the following ways:

- **Direct bag representation retrieval.** As a baseline method, we propose an approach to augment bag features by directly retrieving the nearest neighbor. The setup follows the initial steps of Algo. 1, from line 1 to line 3. Instead of computing the instance correlations, we directly compute the $l_2$ distance between bag representations, formulated as $d_{l_2}(\boldsymbol{H}_n, \tilde{\boldsymbol{H}}_m) = \|\boldsymbol{z}_n - \tilde{\boldsymbol{z}}_m\|_2^2$. For the $n$-th bag representation, we select the nearest neighbor from the retrieval set: $\tilde{\boldsymbol{H}}_{m^*} = \arg\min_{\tilde{\boldsymbol{H}}_m \in \mathcal{H}_r} d_{l_2}(\boldsymbol{H}_n, \tilde{\boldsymbol{H}}_m)$. The merged bag representation $\hat{\boldsymbol{z}}_n = \pi(\boldsymbol{z}_n, \boldsymbol{z}_{m^*})$ is subsequently employed for bag classification, where $m^* = \mathrm{getIndex}(\tilde{\boldsymbol{H}}_{m^*})$.
- **Approximate OT retrieval.** Recall that prior to applying the Sinkhorn's algorithm to solve OT, it is necessary to compute the cost matrix $C = [c(\boldsymbol{h}_i, \tilde{\boldsymbol{h}}_j)]_{ij} = \|\boldsymbol{h}_i - \tilde{\boldsymbol{h}}_j\|_2^2$, where each element is the squared $l_2$ distance between instance representations. To highlight the significance of solving the OT problem, we introduce three simple approximations of OT by directly employing the minimum, average and maximum values of the cost matrix. The $d_{\mathrm{Cmin}}(\boldsymbol{H}_n, \tilde{\boldsymbol{H}}_m) = \min_{ij}[c(\boldsymbol{h}_i, \tilde{\boldsymbol{h}}_j)]_{ij}$ focuses on the closest and most similar instances between bags. The $d_{\mathrm{Cavg}}(\boldsymbol{H}_n, \tilde{\boldsymbol{H}}_m) = \mathrm{average}_{ij}[c(\boldsymbol{h}_i, \tilde{\boldsymbol{h}}_j)]_{ij}$ offers a holistic measure of bag similarity but might lose some finer details. The $d_{\mathrm{Cmax}}(\boldsymbol{H}_n, \tilde{\boldsymbol{H}}_m) = \max_{ij}[c(\boldsymbol{h}_i, \tilde{\boldsymbol{h}}_j)]_{ij}$ focuses on the most dissimilar instances between the two bags. These three measures can be considered as OT approximations that use information from the squared $l_2$ cost matrix.
- **Hausdorff distance retrieval.** The symmetric Hausdorff distance is employed for retrieval, computed as $d_{\mathrm{H}}(\boldsymbol{H}_n, \tilde{\boldsymbol{H}}_m) = \max\{d_{\mathrm{h}}(\boldsymbol{H}_n, \tilde{\boldsymbol{H}}_m), d_{\mathrm{h}}(\tilde{\boldsymbol{H}}_m, \boldsymbol{H}_n)\}$. Here, $d_{\mathrm{h}}(\boldsymbol{H}_n, \tilde{\boldsymbol{H}}_m) = \max_{\boldsymbol{h}_i \in \boldsymbol{H}_n} \min_{\tilde{\boldsymbol{h}}_j \in \tilde{\boldsymbol{H}}_m} \|\boldsymbol{h}_i - \tilde{\boldsymbol{h}}_j\|_2^2$ is the directed Hausdorff distance. The Hausdorff distance is an appropriate retrieval metric when assessing the maximum dissimilarity between two sets, emphasizing the instances that differ the most between the sets. In comparison, OT provides a more comprehensive and versatile choice for retrieval tasks, given its ability to consider all possible matchings between elements in the sets and finds an optimal one.

### 5.1 Classification of Whole Slide Image

**Experimental setup:** In this section, we validate the methodology within both in-domain and out-of-domain contexts. In both contexts, the training set is from Camelyon16. In the in-domain setting, we evaluate our models using the Camelyon16 dataset, where the combination of training and validation sets is treated as $\mathcal{D}_r$, which is denoted as *Retr$_I$*. For the out-of-domain context, we use Camelyon16 as $\mathcal{D}_o$. As for $\mathcal{D}_r$, we consider two configurations: 1) retrieval from the training and validation sets of both Camelyon16 and Camelyon17, and 2) retrieval solely from the training

Table 2: Results on CPTAC-UCEC in domain classification, domain adaptation from CPTAC-LSCC to CPTAC-UCEC.

|  | In Domain | | Out-of-Domain | |
|  | AUC | Accuracy | AUC | Accuracy |
|---|---|---|---|---|
| CLAM | 0.9500 | 0.9285 | 0.4986 | 0.6521 |
| RAM-MIL | **0.9667** | **0.9382** | **0.6056** | **0.6526** |

Table 3: Results on TCGA-NSCLC (subtyping).

| Method | AUC | Accuracy |
|---|---|---|
| CLAM | 0.9420 | 0.8640 |
| Scaling ViT | 0.9516 | 0.8821 |
| TransMIL | **0.9603** | 0.8835 |
| Bayes-MIL | 0.9451 | 0.8965 |
| RAM-MIL | 0.9457 | **0.8988** |

Table 4: Tumor stage classification results on CAMELYON17.

| Method | AUC | Accuracy |
|---|---|---|
| CLAM | 0.7803 | 0.60 |
| Bayes-MIL | 0.8070 | 0.64 |
| RAM-MIL | **0.8139** | **0.65** |

and validation sets of Camelyon17, referred to as $Retr_{IO}$ and $Retr_O$, respectively. In the Camelyon17 challenge, each slide is labeled as either isolated tumour cells (ITC), micro-met, macro-met, or negative. ITC are strictly defined as single tumor cells or clusters smaller than 0.2 mm or less than 200 cells [44]. Following the principles of MIL, we treated ITC as the positive class. More results about treating ITC as the negative class are shown in the supplemental. To ensure robustness and avoid the influence of outliers, each experiment is executed 10 times on randomly partitioned train/validation/test sets.

The baseline model we use for comparison is CLAM, an extension of attention-based Multiple Instance Learning (MIL) that incorporates an instance clustering loss. In our case, the pre-trained model is a CLAM model for binary classification. We also make comparisons with other models such as ABMIL, DSMIL, TransMIL, Bayes-MIL, HHOT, and Mixup. For the ablation study, we add our proposed $l_2$ retrieval method with direct bag representation retrieval. For both the Mixup and $l_2$ retrieval models, we ensure that the source of the augmented data aligns with our in-domain and out-of-domain settings. For the OT-based pathology work, HHOT focuses on comparing whole slide images using uniform weight-based OT and employs kNN as a discriminative method. Note that Mixup and our proposed methods use CLAM as the MIL method. Furthermore, Mixup shares the same merging function with our retrieval method, which is a simple addition or a convex combination, in the reported experiments. Our method does not need the ground-truth label information for retrieval set while Mixup uses that for the in-domain experiments. More results about the merging function is shown in the supplemental.

When we compute the RAM-MIL, we often deal with very large WSIs that can contain over 100,000 instances. This makes the calculation of OT time-consuming. To speed things up, we first perform pre-selection of potentially important instances. Instead of using all the instances in a bag $\mathcal{H}_n$, we pick the top 20% of instances based on their attention scores and normalized the attention scores. We provide more detailed comparisons in percentage of instances used, in the supplemental. In this way, we are focusing on the most important instances and making the whole process more efficient.

**In-domain comparisons:** Table 1 presents the in-domain results. Simply using a uniform weight-based Optimal Transport (OT) in HHOT-KNN fails to deliver satisfactory performance. Mixup, $l_2$ retrieval, and RAM-MIL all enhance the classification performance for in-domain data beyond the CLAM baseline. The straightforward $l_2$ retrieval-based MIL outpaces Mixup, demonstrating that selecting the nearest neighbor using a basic metric offers superior augmentation compared to random data selection. Using OT as a distance metric, the retrieval-based MIL method surpasses all the other compared methods for the in-domain dataset. Notably, the $Retr_{IO}$ configuration, which employs the most extensive data set for retrieval (CAMELYON16 and CAMELYON17), offers the most significant boost in in-domain classification accuracy. Table 2 shows RAM-MIL consistently presents higher AUC and accuracy on CPTAC-UCEC. We also explore the performance of subtypes classification in Table 3, showing that RAM-MIL outperforms existing methods in accuracy, while providing AUC higher than the CLAM baseline. Compared with the results on tumor stage classification, our method enhances the model performance as shown in Table 4.

**Out-of-domain comparisons:** The right-hand side of Table 1 showcases the results for the out-of-domain setting. Notably, all retrieval methods, including ours, enhance the classification performance

Table 5: Ablation study evaluated on CAMELYON16 and CAMELYON17.

| | In-Domain (CAM16) | | Out-of-Domain (CAM17) | |
|---|---|---|---|---|
| | AUC | Accuracy | AUC | Accuracy |
| Approx-OT-min $Retr_I$ | 0.9301±0.033 | 0.8800±0.043 | - | - |
| Approx-OT-min $Retr_{IO}$ | 0.8601±0.080 | 0.8000±0.062 | 0.7640±0.058 | 0.7353±0.043 |
| Approx-OT-min $Retr_O$ | 0.9196±0.063 | 0.8750±0.056 | 0.7680±0.065 | 0.7274±0.046 |
| Approx-OT-avg $Retr_I$ | 0.9227±0.038 | 0.8600±0.036 | - | - |
| Approx-OT-avg $Retr_{IO}$ | 0.8865±0.063 | 0.8425±0.061 | 0.7625±0.060 | 0.7174±0.041 |
| Approx-OT-avg $Retr_O$ | 0.9195±0.063 | 0.8750±0.056 | 0.7641±0.054 | 0.7374±0.061 |
| Approx-OT-max $Retr_I$ | 0.9174±0.049 | 0.8625±0.044 | - | - |
| Approx-OT-max $Retr_{IO}$ | 0.9182±0.047 | 0.8650±0.048 | 0.7576±0.060 | 0.7214±0.062 |
| Approx-OT-max $Retr_O$ | 0.9195±0.063 | 0.8750±0.056 | 0.7681±0.065 | 0.7414±0.038 |
| $l_2$ $Retr_I$ | 0.9281±0.047 | 0.8800±0.055 | - | - |
| $l_2$ $Retr_{IO}$ | 0.9241±0.048 | 0.8950±0.048 | 0.7627±0.055 | 0.7353±0.051 |
| $l_2$ $Retr_O$ | 0.9398±0.045 | 0.8950±0.055 | 0.7738±0.050 | 0.7493±0.050 |
| Hausdorff $Retr_I$ | 0.9273±0.057 | 0.8850±0.049 | - | - |
| Hausdorff $Retr_{IO}$ | 0.9226±0.055 | 0.8725±0.060 | 0.7651±0.054 | 0.7434±0.051 |
| Hausdorff $Retr_O$ | 0.9322±0.046 | 0.8950±0.048 | 0.7695±0.056 | 0.7513±0.051 |
| RAM-MIL $Retr_I$ | **0.9451±0.036** | 0.8925±0.050 | - | - |
| RAM-MIL $Retr_{IO}$ | 0.9365±0.052 | **0.9200±0.050** | **0.7974±0.054** | 0.7433±0.073 |
| RAM-MIL $Retr_O$ | 0.9419±0.048 | 0.9175±0.051 | 0.7681±0.058 | **0.7795±0.021** |

for out-of-domain data, without accessing its labels. This suggests that retrieval benefits from the transfer of representative information from the source to the target domain via the merging of bag features. The $l_2$ retrieval, while trailing Mixup in terms of accuracy, surpasses it in terms of AUC. Our OT-based retrieval achieves the highest performance across both AUC and accuracy metrics. Table 2 presents the results on domain adaptation from CPTAC-UCEC to CPTAC-LSCC. RAM-MIL demonstrates a notable improvement of 0.1065 in AUC compared to CLAM. The underlying reason for the good performance might be it could consider all possible matchings between elements in the sets and finds an optimal one.

**Ablation study** Table 5 presents the ablation study using different retrieval methods, which shows that even $l_2$ retrieval, which operates without considering instance-level distances, surpasses the performance of the three OT approximation methods in both in-domain and out-of-domain settings. Among these, only the Approx-OT-min, when used for in-domain data retrieval, achieves a performance comparable to that of $l_2$ retrieval. This underscores the fact that relying on simple statistics of the cost matrix cannot provide a robust and consistent approximation of the optimal transport. This outcome reinforces the necessity of solving the complete optimal transport problem, using Sinkhorn's algorithm, to attain reliable results.

## 5.2 Dimensionality Reduction

We next show that retrieval-based methods can effectively reduce the intrinsic dimension. Figure 3 shows the intrinsic dimension for various methods. The proposed RAM-MIL based on optimal transport delivers the lowest intrinsic dimension, among all methods, both retrieval and vanilla models. Interestingly, Mixup seems to increase the intrinsic dimension, suggesting that random selection and merging of bag representations do not necessarily lead to a more compact representation. This could explain why Mixup only offers marginal improvements.

Figure 3 shows that while the CLAM model has an intrinsic dimension between 5-7, RAM-MIL achieves a dimension between 2-4. Based on these observations, we choose dimension 5 to guide manual dimensionality reduction. The objectives of this exercise are two-fold: firstly, to validate that intrinsic dimension can accurately explain performance, and secondly, to illustrate that simple methods like SVD are not necessarily beneficial for dimensionality reduction of bag representations.

In our experiment, we employ SVD to reduce each bag representation of CLAM and each merged bag representation of RAM-MIL to a 5-dimensional vector. Then, we use these vectors and the bag labels to train a single logistic regression classifier. The results indicate that RAM-MIL significantly outperforms CLAM in classification accuracy. This experiment shows that a simple dimensionality reduction method like SVD does not significantly improve the performance of the CLAM baseline.

Table 6: Classification on CAMELYON16 with bag representations of reduced dimensions (5-dim).

| | In-Domain (CAM16) | | Out-of-Domain (CAM17) | |
| --- | --- | --- | --- | --- |
| | AUC | Accuracy | AUC | Accuracy |
| CLAM | 0.9216±0.057 | 0.6625±0.177 | 0.7486±0.040 | 0.4991±0.105 |
| RAM-MIL | **0.9273±0.072** | **0.8475±0.157** | **0.7509±0.074** | **0.7053±0.113** |

However, RAM-MIL, which organizes the latent space by merging nearest neighbor features and then applies SVD, can achieve a commendable performance.

## 5.3 Interpretability with OT for MIL

We show that by using optimal transport retrieval, a novel method for highlighting the important instances could be derived. For a pair of source bag $\boldsymbol{H}$ and target bag $\tilde{\boldsymbol{H}}$, solving the optimal transport problem generates the transport matrix $\boldsymbol{T} = [T_{ij}]_{ij}$. Suppose that we are interested in the instance labels of the source bag, and suppose that the instance labels of the target bag are known. The positive instances in the source bag are simply the instances transported to the positive instances in the target bag, i.e., $\hat{y}_i = \tilde{y}_{j^* = \arg\max_j T_{ij}}$. Here, $\hat{y}$ is the predicted instance label for the source bag, and and $\tilde{y}$ is the true instance label in the target bag. Figure 4 shows this method could provide a good coverage of the positive area, without model training or accessing the bag label. In contrast to OT, retrieval-based methods utilizing other distance metrics do not possess this specific property.

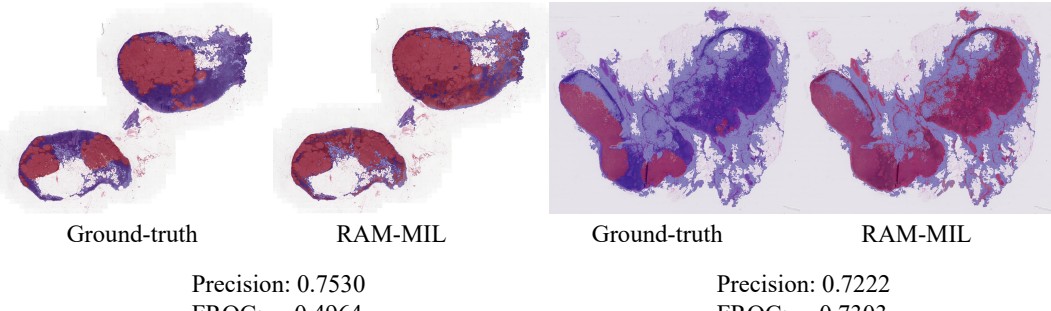

| Ground-truth | RAM-MIL | Ground-truth | RAM-MIL |

Precision: 0.7530
FROC:  0.4964

Precision: 0.7222
FROC:  0.7303

Figure 4: The visualization of predicted positive instances (red regions) in whole slide images for RAM-MIL. FROC is defined as the average sensitivity (recall) at 6 predefined false positive (FP) rates: 1/4, 1/2, 1, 2, 4 and 8 FPs. For reference, the averaged precision and FROC in Bayes-MIL [11] (one of the SOTA method designed for instance-level performance) are 0.8107 and 0.4919.

## 6 Conclusion

In this work, we investigate the out-of-domain performance of multiple instance learning model, which is crucial for the ML-assisted medical diagnosis with the whole slide images. An optimal transport based retrieval method is proposed based on the understanding that there exists a negative correlation between MIL performance and input data dimension. The proposed RAM-MIL framework works for both in-domain and out-of-domain, meanwhile outperforms state-of-the-art MIL methods. We hypothesize that OT-based retrieval could reduce the intrinsic dimension therefore improves the MIL performance and validate the hypothesis by computing the intrinsic dimension of bag representations. The transportation matrix of OT distance provides a tool to visualize the correspondence between original and retrieval instances and makes our retrieval method interpretable. A limitation is the efficiency of optimal transport solver is constrained by the number of instances within a bag. Although we have mitigated this problem by selecting partial instances based on high attention values, we anticipate that a more principled method will be sought in future works.

## Acknowledgements

This work was supported by a grant from the Research Grants Council of the Hong Kong Special Administrative Region, China (Project No. CityU 11215820), and a grant from Guangzhou Bingli Technology Co., Ltd.

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

# A Proof

*Proof.* Since the bag of instances $\boldsymbol{X}$ is sampled from the probability distribution $\mu(\boldsymbol{x})$, we have the upper bound for Wasserstein distance between $\boldsymbol{X}$ and $\mu$ [45],

$$\mathbb{E}[\mathcal{W}_p(\boldsymbol{X}, \mu)] \leq K^{-\frac{1}{d_\mu}}, \tag{6}$$

where $K$ is the number of samples. Next we define the function $\sigma(\boldsymbol{x})$ as a small perturbation function $\sigma(\boldsymbol{x}) = \boldsymbol{x} + \boldsymbol{\delta}$ and let $\tilde{\boldsymbol{X}} = \{\sigma(\boldsymbol{x}) : \boldsymbol{x} \in \boldsymbol{X}\}$. Using the triangle inequality, we have

$$\mathbb{E}[\mathcal{W}_p(\boldsymbol{X}, \tilde{\boldsymbol{X}})] \leq \mathbb{E}[\mathcal{W}_p(\boldsymbol{X}, \mu)] + \mathbb{E}[\mathcal{W}_p(\tilde{\boldsymbol{X}}, \mu)] \leq 2K^{-\frac{1}{d_\mu}} + C, \tag{7}$$

where $C$ is a constant as a result of the perturbation. As the function $S(\cdot)$ is Lipschitz continuous, we have

$$|S(\boldsymbol{X}) - S(\tilde{\boldsymbol{X}})| \leq L \cdot \mathbb{E}[\mathcal{W}_p(\boldsymbol{X}, \tilde{\boldsymbol{X}})] \leq O(L \cdot K^{-\frac{1}{d_\mu}}). \tag{8}$$

Similar to TransMIL [5], let $\Phi : \mathcal{X} \to \mathbb{R}^n$ be any invertible map, where its inverse mapping is expressed as $\Phi^{-1} : \mathbb{R}^d \to \mathcal{X}$. Then we have:

$$S(\Phi^{-1}(\Phi_{\boldsymbol{X} \in \mathcal{X}}(\{\sigma(\boldsymbol{x}) : \boldsymbol{x} \in \boldsymbol{X}\}))) = S(\Phi^{-1}(\Phi_{\tilde{\boldsymbol{X}} \in \mathcal{X}}(\tilde{\boldsymbol{X}}))) = S(\tilde{\boldsymbol{X}}). \tag{9}$$

Let $\gamma = S \circ \Phi^{-1}$. As $|S(\boldsymbol{X}) - S(\tilde{\boldsymbol{X}})| \leq O(L \cdot K^{-\frac{1}{d_\mu}})$, we have

$$|S(\boldsymbol{X}) - \gamma(\Phi_{\boldsymbol{X} \in \mathcal{X}}\{\sigma(\boldsymbol{x}) : \boldsymbol{x} \in \boldsymbol{X}\})| \leq O(L \cdot K^{-\frac{1}{d_\mu}}). \tag{10}$$

$\square$

In this proof, the transformation $\Phi(\cdot) = \mathcal{A}(\cdot)$. This proof could be easily extended to the representations $\boldsymbol{H}$ by assuming a probability measure over the instance representations $\boldsymbol{h}$ and replacing $\boldsymbol{X}$ with $\boldsymbol{H}$. In this case, the transformation $\Phi(\boldsymbol{H}) = \sum_{k=1}^{K} a_k \boldsymbol{h}_k$.

Table 7: Results on general MIL datasets. Experiments were run 5 times and the average classification accuracy ($\pm$ a standard error of a mean) is reported.

| Method | MUSK1 | MUSK2 | FOX | TIGER | ELEPHANT |
|---|---|---|---|---|---|
| Attention | 0.892±0.090 | 0.858±0.106 | 0.615±0.096 | 0.839±0.054 | 0.868±0.054 |
| Attention-Gated | 0.900±0.088 | 0.863±0.094 | 0.603±0.068 | **0.845±0.046** | 0.857±0.064 |
| CLAM | 0.900±0.136 | 0.860±0.128 | 0.610±0.128 | 0.805±0.052 | 0.860±0.080 |
| RAM-MIL | **0.911±0.130** | **0.870±0.142** | **0.645±0.117** | 0.820±0.040 | **0.879±0.096** |

# B General MIL dataset

Table 7 presents the performance of RAM-MIL on general MIL datasets [46, 47], offering a comparison with baseline methods. The results indicate that OT-based retrieval generally enhances the classification performance. The sole exception is observed with the TIGER dataset, where both CLAM and RAM-MIL are outperformed. This discrepancy might be attributed to CLAM, as RAM-MIL uses CLAM as a pretrained model for attention weights and bag representation extraction. Nonetheless, RAM-MIL still improves over its CLAM baseline on TIGER. Note that our primary focus lies on the more challenging WSI datasets, hence our models are not extensively optimized for general datasets. The data in these general datasets are typically of lower dimensionality and present less challenging conditions. Therefore, any potential underperformance in these contexts should not detract from the strength of our models in handling the WSI data.

# C Experiment Details of WSI Classification

We present the experimental details, ablation studies and analysis step-by-step.

**MIL Pre-training.** For the backbone MIL model we use the the same parameter setup as CLAM. The model parameters are updated via the Adam optimizer with an L2 weight decay of 1e-5 and a learning rate of 2e-4. Each result is obtained with 10-fold splits of training/validation/testing sets.

Table 8: Ablation study for the percentage of instances used on CAMELYON16 and CAMELYON17.

| | In-Domain (CAM16) | | Out-of-Domain (CAM17) | |
| --- | --- | --- | --- | --- |
| | AUC | Accuracy | AUC | Accuracy |
| 10% attention $\text{Retr}_\text{I}$ | 0.9440±0.037 | 0.8975±0.052 | - | - |
| 10% attention $\text{Retr}_\text{IO}$ | 0.9365±0.052 | **0.9200±0.050** | **0.7974±0.054** | 0.7433±0.073 |
| 10% attention $\text{Retr}_\text{O}$ | 0.9414±0.046 | 0.8975±0.056 | 0.7775±0.050 | 0.7392±0.063 |
| 20% attention $\text{Retr}_\text{I}$ | **0.9451±0.036** | 0.8925±0.050 | - | - |
| 20% attention $\text{Retr}_\text{IO}$ | 0.9341±0.051 | 0.8925±0.053 | 0.7651±0.056 | 0.7714±0.030 |
| 20% attention $\text{Retr}_\text{O}$ | 0.9419±0.048 | 0.9175±0.051 | 0.7681±0.058 | **0.7795±0.021** |

**Neighbor Selection.** After pre-training the MIL model, we obtain the slide-level feature and the attention scores predicted by the network. As computing the optimal transport distance based on all instances is time-consuming, we approximate the distance with a part of samples in a bag.

$$d_{\text{OT}}(\mu, \nu) = \min_{\boldsymbol{T} \in \mathcal{T}(\boldsymbol{\alpha}, \tilde{\boldsymbol{\alpha}})} \sum_{i=1}^{|\boldsymbol{\alpha}|} \sum_{j=1}^{|\tilde{\boldsymbol{\alpha}}|} c(\boldsymbol{h}_i, \tilde{\boldsymbol{h}}_j) T_{ij} + \beta \cdot \sum_{ij} T_{ij} \log T_{ij} \tag{11}$$

$$s.t., \boldsymbol{T}^\top \boldsymbol{1}_K = \boldsymbol{\alpha}, \boldsymbol{T} \boldsymbol{1}_{\tilde{K}} = \tilde{\boldsymbol{\alpha}}, \boldsymbol{T} \geq 0$$

where $\boldsymbol{\alpha}$ and $\tilde{\boldsymbol{\alpha}}$ are the new attention vector obtained by selecting top $\eta\%$ from $\boldsymbol{a}$ and $\tilde{\boldsymbol{a}}$. In other words, we approximate a bag with $\eta\%$ of instances with the highest attention values generated by pretrained MIL model. As shown in Table 8, we set $\eta = 10$ and $\eta = 20$ for the ablation study. In this experiment, we use Regularization term of 0.5 and Max number of iterations 1000.

It is observed from Table 8, improving the percentage of data improves the performance on $\text{Retr}_\text{I}$ and $\text{Retr}_\text{O}$. On $\text{Retr}_\text{IO}$, the performance is saturated when using only 10% of all patches. Differentiating in-domain and out-of-domain data for retrieval could be easily accomplished by representing a bag by a few amount of instances.

Table 9: Ablation study for different merge functions on CAMELYON16 and CAMELYON17.

| | In-Domain (CAM16) | | Out-of-Domain (CAM17) | |
| --- | --- | --- | --- | --- |
| | AUC | Accuracy | AUC | Accuracy |
| $\text{Merge}_{add}$(2-feats) $\text{Retr}_\text{I}$ | 0.9409±0.038 | 0.9000±0.049 | - | - |
| $\text{Merge}_{add}$(2-feats) $\text{Retr}_\text{IO}$ | 0.9341±0.051 | 0.8925±0.053 | 0.7651±0.056 | 0.7714±0.030 |
| $\text{Merge}_{add}$(2-feats) $\text{Retr}_\text{O}$ | 0.9414±0.046 | 0.8975±0.056 | 0.7775±0.050 | 0.7392±0.063 |
| $\text{Merge}_{add}$(3-feats) $\text{Retr}_\text{I}$ | 0.9383±0.050 | 0.9175±0.051 | - | - |
| $\text{Merge}_{add}$(3-feats) $\text{Retr}_\text{IO}$ | 0.9313±0.044 | 0.9000±0.045 | 0.7641±0.059 | 0.7553±0.043 |
| $\text{Merge}_{add}$(3-feats) $\text{Retr}_\text{O}$ | 0.9391±0.051 | 0.9175±0.045 | 0.7644±0.059 | 0.7754±0.022 |
| $\text{Merge}_{convex}$(2-feats) $\text{Retr}_\text{I}$ | 0.9451±0.036 | 0.8925±0.050 | - | - |
| $\text{Merge}_{convex}$(2-feats) $\text{Retr}_\text{IO}$ | 0.9365±0.052 | 0.9200±0.050 | 0.7974±0.054 | 0.7433±0.073 |
| $\text{Merge}_{convex}$(2-feats) $\text{Retr}_\text{O}$ | 0.9419±0.048 | 0.9175±0.051 | 0.7681±0.058 | 0.7795±0.021 |
| $\text{Merge}_{convex}$(3-feats) $\text{Retr}_\text{I}$ | 0.9398±0.043 | 0.8975±0.052 | - | - |
| $\text{Merge}_{convex}$(3-feats) $\text{Retr}_\text{IO}$ | 0.9435±0.038 | 0.8975±0.052 | 0.7652±0.052 | 0.7714±0.030 |
| $\text{Merge}_{convex}$(3-feats) $\text{Retr}_\text{O}$ | 0.9417±0.048 | 0.9050±0.050 | 0.7690±0.056 | 0.7755±0.021 |

**Merge Function.** Table 9 presents the results using different merge functions. To generate the bag representations, we employ two merge functions: 1) simple addition, referred to as $\text{Merge}_{add}$; 2) convex combination, referred to as $\text{Merge}_{convex}$. Additionally, '2-feats' and '3-feats' refer to bag representation that are merged with 1 nearest neighbor or 2 nearest neighbors, respectively. For convex combination, '2-feats' uses coefficients of $0.6$ and $0.4$, while '3-feats' uses coefficients of $0.6, 0.2$ and $0.2$, where the greatest coefficient corresponds to the original representation. This experiment is done with $\eta = 10$.

It is derived from Table 9 that using 1 nearest neighbor and convex combination presents the best performance. Using 2 nearest neighbors and addition presents the similar results.

**Classification Training.** Finally, we train a single logistic regression classifier using the merged representation. The Adam optimizer is used to update the model parameters, with a L2 weight decay of 1e-4 and a learning rate of 2e-4. The models are trained for a minimum of 40 epochs and up to a maximum of 200 epochs if the early stopping criterion is not met. This criterion involves monitoring the validation loss each epoch and if it has not decreased from the previous low for over 15 consecutive epochs, early stopping is used.

## D    Patch-level results on tumor slides of CAMELYON16.

Table 10: Patch-level results on tumor slides of CAMELYON16.

|  | P-Prec.($\uparrow$) | FROC($\uparrow$) |
|---|---|---|
| DSMIL | 0.1030 | 0.4443 |
| CLAM | 0.6068 | 0.4792 |
| TransMIL | 0.1726 | 0.4797 |
| Bayes-MIL | **0.8107** | 0.4919 |
| RAM-MIL | 0.6114 | **0.5281** |

The Tumor-Precision is calculated by the precision of classifying the tumor patches. The Patch-Precision is calculated by averaging the precision of classifying both normal and tumor patches. The Patch-FROC is defined as the average sensitivity (recall) at 6 predefined false positive rates: 1/4, 1/2, 1, 2, 4 and 8 FPs per WSI.

In Table 17, RAM-MIL presents the second best precision and the best FROC on the patch-level segmentation. This indicates that using transport matrix for interpreting the patch-level classification achieves the best overall performance in the trade-off of false positive rate and recall. By contrast, Bayes-MIL could only obtain a high precision, which reduces the number of false alarm. However, for the application of medical WSI, reducing false negative (better recall and FROC) is supposed to be more important as classifying a positive instance to be negative is unacceptable in the application of medical pregnosis or diagnosis.

## E    Results on CAMELYON17.

A general setting is to treat ITC as the negative class in the CAMELYON17 dataset.

Table 11: Results on CAMELYON16 and CAMELYON17 for in-domain classification and unsupervised domain adaptation, for comparisons with related methods.

|  | In-Domain (CAM16) | | Out-of-Domain (CAM17) | |
|---|---|---|---|---|
|  | AUC | Accuracy | AUC | Accuracy |
| CLAM | 0.9177±0.044 | 0.8650±0.060 | 0.8399±0.059 | 0.7796±0.099 |
| Mixup $\text{Retr}_\text{I}$ | 0.9260±0.051 | 0.8825±0.051 | - | - |
| Mixup $\text{Retr}_\text{IO}$ | 0.9271±0.048 | 0.8850±0.046 | 0.8418±0.054 | 0.8016±0.059 |
| Mixup $\text{Retr}_\text{O}$ | 0.9271±0.045 | 0.8825±0.048 | 0.8397±0.057 | 0.8095±0.063 |
| $l_2$ $\text{Retr}_\text{I}$ | 0.9281±0.047 | 0.8800±0.055 | - | - |
| $l_2$ $\text{Retr}_\text{IO}$ | 0.9241±0.048 | 0.8950±0.048 | 0.8403±0.049 | 0.7914±0.052 |
| $l_2$ $\text{Retr}_\text{O}$ | 0.9398±0.045 | 0.8950±0.055 | 0.8412±0.055 | 0.7998±0.056 |
| RAM-MIL $\text{Retr}_\text{I}$ | **0.9451±0.036** | 0.8925±0.050 | - | - |
| RAM-MIL $\text{Retr}_\text{IO}$ | 0.9365±0.052 | **0.9200±0.050** | **0.8475±0.055** | 0.8236±0.051 |
| RAM-MIL $\text{Retr}_\text{O}$ | 0.9419±0.048 | 0.9175±0.051 | 0.8413±0.053 | **0.8457±0.027** |

Table 12: Ablation study evaluated on CAMELYON16 and CAMELYON17.

| | In-Domain (CAM16) | | Out-of-Domain (CAM17) | |
|---|---|---|---|---|
| | AUC | Accuracy | AUC | Accuracy |
| Approx-OT-min Retr$_I$ | 0.9301±0.033 | 0.8800±0.043 | - | - |
| Approx-OT-min Retr$_{IO}$ | 0.8601±0.080 | 0.8000±0.062 | 0.8380±0.056 | 0.7935±0.065 |
| Approx-OT-min Retr$_O$ | 0.9196±0.063 | 0.8750±0.056 | 0.8397±0.056 | 0.7915±0.048 |
| Approx-OT-avg Retr$_I$ | 0.9227±0.038 | 0.8600±0.036 | - | - |
| Approx-OT-avg Retr$_{IO}$ | 0.8865±0.063 | 0.8425±0.061 | 0.8401±0.056 | 0.7902±0.051 |
| Approx-OT-avg Retr$_O$ | 0.9195±0.063 | 0.8750±0.056 | 0.8412±0.053 | 0.7823±0.063 |
| Approx-OT-max Retr$_I$ | 0.9174±0.049 | 0.8625±0.044 | - | - |
| Approx-OT-max Retr$_{IO}$ | 0.9182±0.047 | 0.8650±0.048 | 0.8406±0.056 | 0.7854±0.052 |
| Approx-OT-max Retr$_O$ | 0.9195±0.063 | 0.8750±0.056 | 0.8412±0.063 | 0.7925±0.058 |
| $l_2$ Retr$_I$ | 0.9281±0.047 | 0.8800±0.055 | - | - |
| $l_2$ Retr$_{IO}$ | 0.9241±0.048 | 0.8950±0.048 | 0.8403±0.049 | 0.7914±0.052 |
| $l_2$ Retr$_O$ | 0.9398±0.045 | 0.8950±0.055 | 0.8412±0.055 | 0.7998±0.056 |
| Hausdorff Retr$_I$ | 0.9273±0.057 | 0.8850±0.049 | - | - |
| Hausdorff Retr$_{IO}$ | 0.9226±0.055 | 0.8725±0.060 | 0.8403±0.056 | 0.7932±0.053 |
| Hausdorff Retr$_O$ | 0.9322±0.046 | 0.8950±0.048 | 0.8395±0.063 | 0.8000±0.061 |
| RAM-MIL Retr$_I$ | **0.9451±0.036** | 0.8925±0.050 | - | - |
| RAM-MIL Retr$_{IO}$ | 0.9365±0.052 | **0.9200±0.050** | **0.8475±0.055** | 0.8236±0.051 |
| RAM-MIL Retr$_O$ | 0.9419±0.048 | 0.9175±0.051 | 0.8413±0.053 | **0.8457±0.027** |

Table 13: Ablation study for the percentage of instances used on CAMELYON16 and CAMELYON17.

| | In-Domain (CAM16) | | Out-of-Domain (CAM17) | |
|---|---|---|---|---|
| | AUC | Accuracy | AUC | Accuracy |
| 10% attention Retr$_I$ | 0.9440±0.037 | 0.8975±0.052 | - | - |
| 10% attention Retr$_{IO}$ | 0.9365±0.052 | **0.9200±0.050** | **0.8475±0.055** | 0.8236±0.051 |
| 10% attention Retr$_O$ | 0.9414±0.046 | 0.8975±0.056 | 0.8466±0.047 | 0.8037±0.062 |
| 20% attention Retr$_I$ | **0.9451±0.036** | 0.8925±0.050 | - | - |
| 20% attention Retr$_{IO}$ | 0.9341±0.051 | 0.8925±0.053 | 0.8473±0.052 | 0.8216±0.030 |
| 20% attention Retr$_O$ | 0.9419±0.048 | 0.9175±0.051 | 0.8413±0.053 | **0.8457±0.027** |

Table 14: Ablation study for different merge functions on CAMELYON16 and CAMELYON17.

| | In-Domain (CAM16) | | Out-of-Domain (CAM17) | |
|---|---|---|---|---|
| | AUC | Accuracy | AUC | Accuracy |
| Merge$_{add}$(2-feats) Retr$_I$ | 0.9409±0.038 | 0.9000±0.049 | - | - |
| Merge$_{add}$(2-feats) Retr$_{IO}$ | 0.9341±0.051 | 0.8925±0.053 | 0.8473±0.052 | 0.8216±0.030 |
| Merge$_{add}$(2-feats) Retr$_O$ | 0.9414±0.046 | 0.8975±0.056 | 0.8390±0.056 | 0.7695±0.056 |
| Merge$_{add}$(3-feats) Retr$_I$ | 0.9383±0.050 | 0.9175±0.051 | - | - |
| Merge$_{add}$(3-feats) Retr$_{IO}$ | 0.9313±0.044 | 0.9000±0.045 | 0.8432±0.055 | 0.8176±0.040 |
| Merge$_{add}$(3-feats) Retr$_O$ | 0.9391±0.051 | 0.9175±0.045 | 0.8402±0.056 | 0.8457±0.020 |
| Merge$_{convex}$(2-feats) Retr$_I$ | 0.9451±0.036 | 0.8925±0.050 | - | - |
| Merge$_{convex}$(2-feats) Retr$_{IO}$ | 0.9365±0.052 | 0.9200±0.050 | 0.8475±0.055 | 0.8236±0.051 |
| Merge$_{convex}$(2-feats) Retr$_O$ | 0.9419±0.048 | 0.9175±0.051 | 0.8413±0.053 | 0.8457±0.027 |
| Merge$_{convex}$(3-feats) Retr$_I$ | 0.9398±0.043 | 0.8975±0.052 | - | - |
| Merge$_{convex}$(3-feats) Retr$_{IO}$ | 0.9435±0.038 | 0.8975±0.052 | 0.8473±0.049 | 0.7755±0.067 |
| Merge$_{convex}$(3-feats) Retr$_O$ | 0.9417±0.048 | 0.9050±0.050 | 0.8474±0.047 | 0.8097±0.057 |

# F  Comparisons with domain adaptation methods

There have been a large volume of research studying the domain adaptation in patch-level models, categorized as staining transfer and domain adversarial learning.

Patch-level staining transfer: Cho et al. [31], Shaban et al. [32] and Zanjani et al. [33] use generative adversarial networks (GAN) to learn the staining difference implicitly. Images are transferred to the target domain, by the trained generators in GAN. A simple neural network is then fit on the generated images for discriminative tasks.

Patch-level domain adversarial learning: Lafarge et al. [34] and Ciga et al. [35] apply gradient reversal method for adapting the model for discriminative tasks in medical images. Ren et al. [36] uses Siamese network in domain adversarial learning. Brieu et al. [37], Kapil et al. [38] and Gadermayr et al. [39] for transfer images between domains for segmentation tasks. These methods are not directly comparable with our methods, as they require patch-level labels during training, which is not realistic in real-world scenario. In experiment, we adapt Shaban et al. [32] with least modification, to compare with the retrieval based multiple instance learning. We run Shaban et al. [32] in patch-level in the Camelyon17 domain adaptation setting. For obtaining the WSI features, we use averaged pooling. For fairness, the same classifier as RAM-MIL is used for Shaban et al. [32]. Here are the results:

Table 15: Comparisons with patch-level domain adaptation methods

|  | AUC(↑) | Accuracy(↑) |
| --- | --- | --- |
| Shaban et al. [32] | 0.6458 | 0.6889 |
| RAM-MIL | 0.8209 | 0.7667 |

Slide-level methods: Yang et al. [48] propose a slide-level domain adaptation method that involves local and global adversarial loss to train a weakly supervised learning model. RAM-MIL outperforms this method in the domain adaptation evaluation on Camelyon17.

Table 16: Comparisons with slide-level domain adaptation methods

|  | AUC(↑) | Accuracy(↑) |
| --- | --- | --- |
| Yang et al. [48] | 0.5867 | 0.6786 |
| RAM-MIL | 0.7974 | 0.7795 |

Other methods: Previous methods in slide-level classification [24] and methods in other fields [25] also consider the out-of-domain testing performance. However, they did not provide a method for domain adaptation, while RAM-MIL has. We compare RAM-MIL with several most advanced baselines in slide-level classification in our experiments.

# G  Comparisons with other retrieval methods

For the information retrieval works (Yottixel [40] and SISH [41]) of whole slide image, we will add them into the related works section. However, these two methods are not directly comparable, as they focus on the retrieval problem while our goal is to use retrieval to enhance the weakly supervised classification (MIL). An executable way is to replace the OT-retrieval module in RAM-MIL with these retrieval methods. As there is no available implementation of Yottixel, and SISH is a more advanced retrieval method, we integrate SISH retrieval into the RAM-MIL process. The experiment runs on Camelyon17 (including 5 hospitals), with 2 hospitals as the in-domain dataset and the rest being the out-of-domain dataset. Here are the experimental results:

Table 17: Comparisons with other retrieval methods

|  | AUC(↑) | Accuracy(↑) |
| --- | --- | --- |
| CLAM-SISH | 0.7647 | 0.7444 |
| RAM-MIL | 0.8209 | 0.7667 |

# H   More results on Dimensionality Reduction.

Directly reducing the normal dimension to the estimated intrinsic dimension is a brute-force method to improve the performance. To validate if this argument holds, we use a masked auto-encoder (MAE) with output dimension to be 5 (the estimated intrinsic dimension) to extract the features from patches. To ensure a good quality of extracted features, we fine-tune the MAE with all patches from Camelyon16 for 60 epochs. The performance of CLAM fitted on the extracted features is at the second row of the Table 18:

Table 18: Additional results evaluated on Camelyon16 (in-domain) and Camelyon17 (out-of-domain)

|  | CAMELYON16 | | CAMELYON17 | |
| --- | --- | --- | --- | --- |
|  | AUC | Accuracy | AUC | Accuracy |
| CLAM-SVD-5D | 0.9216 | 0.6625 | 0.8183 | 0.4729 |
| CLAM-MAE-5D | 0.8285 | 0.7368 | 0.7982 | 0.6987 |
| RAM-MIL-SVD-5D | 0.9273 | 0.8475 | 0.8199 | 0.7355 |

