# OpenReview forum: "Retrieval-Augmented Multiple Instance Learning"
_NeurIPS.cc/2023/Conference — NeurIPS 2023 poster_

### Official Review · Reviewer_7vyh · 2023-07-01

**Soundness:** 3 good
**Presentation:** 2 fair
**Contribution:** 3 good
**Rating:** 6
**Confidence:** 4

**Summary:**

This work proposes RAM-MIL: an approach that uses feature alignment in MIL with an optimal transport based method to retrieve nearest bag representations between the train set and target set. This approach shows consistent improvements in ID and OOD settings. Ablation studies demonstrate the relative effectiveness of different ways to compute the nearest neighbors.

**Strengths:**

- The paper brings forth ideas from Unsupervised Domain Adaptation and feature alignment to improve OOD performance in the MIL setting, which is a pressing concern to address the real world deployment of these models. The assumption of using attention values as probability mass values is a logical choice.
- Extensive experimentation and ablations have been conducted to show the effectiveness of the Optimal Transport approach over simpler options like naively calculating nearest neighbors at a bag representation level.  The results show significant improvements in both OOD and ID settings.

**Weaknesses:**

- My main concern is  around the computational cost of inference with RAM-MIL being high due to calculating the distance measures between the train and the retrieval set. Also, in cases where the test set is dynamic, such as one more WSI gets added to the test set, all the pairwise distances between the two sets will need to be recalculated
- The section about interpretability with OT for MIL assumes that instance labels for target bag are known, which is a strong assumption. I would suggest removing this section from the paper and focusing on the OOD generalization part.
- Camelyon-17 has data from 5 different medical centers. It might be worth conducting an experiment where data from a single medical center is used in training, while the other centers are used in retrieval set, since these heldout centers clearly correspond to OOD sets. It is not clear if Camelyon-17 is completely OOD wrt Camelyon-16.

**Questions:**

- The authors mention using top 20% of instances for Camelyon-16 and 17. How much of this choice is a function of the Camelyon datasets where signal is often present in small foci? How will this change in MIL problems like NSCLC and RCC subtyping where the signal is diffused throughout the WSI? [1] has presented some findings on OOD generalization on these 2 datasets.

- The motivation behind the Dimensionality Reduction experiments is unclear. Is the idea to use the reduced dimensions to perform inference in certain time-sensitive applications?

- Is there any additional computational lift for running this method in a multi-class setting?

[1] - SC-MIL: Supervised Contrastive Multiple Instance Learning for Imbalanced Classification in Pathology

**Limitations:**

- The authors mention the limitation of the complexity of the OT approach scaling with the number of instances in the Conclusion section.

---

> ### Author Rebuttal · Authors · 2023-08-10
>
> Thanks for the comments. Here are the responses to weaknesses (W), Questions (Q) and limitations (L).
>
> **Response to W1:**
>
> We report the computation time of RAM-MIL using L2, Hausdorff, approximate-OT and OT for 1 pair of whole slide images as follows. The approximate-OT provides a close performance to full OT while maintaining a low computation cost.
>
> || L2 | Hausdorff | Approximate-OT | Full OT |
> |-------------|---------|---------|---------|---------|
> | Time (s)        | 0.001  | 1.4  | 0.083| 0.498|
>
> > Also, in cases where the test set is dynamic, such as one more WSI gets added to the test set, all the pairwise distances between the two sets will need to be recalculated
>
> This statement is not true. Note that when a new slide is added to the test set, only the pair-wise distances between this slide and the existing retrieval slides (say the number is N) should be calculated, for future usage. We will clarify this in an updated version.
>
> Furthermore, considering a more practical case of streaming data, more efficient approximation can be used for efficiently deploying the streaming inference. For example, a naïve method is to cluster the existing retrieval slides and only compute the distance with existing cluster centers (say the number is C). This will reduce the number of OT-computation by from O(N) to O(C). In this case, OT-computation time would be trivial.
>
> We agree that the dynamic test set is an interesting research topic, but it is out of the scope of this submission. We will leave it as future work.
>
> **Response to W2:**
>
> Thanks for mentioning this typo in Section 5.3. We meant to say “Suppose that we are interested in the instance labels of the target bag, and suppose that the instance labels of the source bag are known.” Switching all “target” and “source” would disambiguate the section 5.3. We will make this revision in the manuscript to make it clear.
>
> **Response to W3:**
>
> We add an experiment that use 2 hospitals of Camelyon17 as the source dataset and 3 other hospitals as the target dataset. Here are the results:
> ||AUC| ACC|
> |-----------|--------|--------|
> |CLAM|0.8155|0.7222|
> |RAM-MIL|**0.8209**|**0.7667**|
>
> **Response to Q1:**
>
> We list the averaged accumulated attention probability values for 10% and 20% patches from Camelyon16, Camelyon17 and TCGA-NSCLC datasets. We observe that NSCLC also present similar signal distribution as Camelyon16 and Camelyon17. We agree with [1] that the imbalanced distributions of positive patches in different WSIs discussed are a crucial issue and will add discussions about [1] to the related work section. The extention of RAM-MIL to the imbalanced positive patch distribution will be an interesting future work.
>
> || C16 | C17 | TCGA-NSCLC |
> |-------------|---------|---------|---------|
> | 10% of patches        | 0.9661  | 0.9667  | 0.9969|
> | 20% of patches    | 0.9882  | 0.9805 | 0.9999|
>
> **Response to Q2:**
>
> Note that the intrinsic dimension is a tool to analyze and validate our methodology. Distinguished from normal dimension, intrinsic dimension is measured by how many dimensions are needed to represent the data without much loss of information. A lower intrinsic dimension would make the learning easier, as the data lies on a less complex manifold or less noisy.
>
> The idea of Section 3.3 - Reduction of Intrinsic Dimension is to show, our proposed retrieval method could reduce the intrinsic dimension of features, thus make the learning of MIL easier. That explains why RAM-MIL has better performance compared with other methods.
>
> The idea of Section 5.2 – Dimensionality Reduction is to show, simply reducing the normal dimension by SVD can have bad performance. Retrieval could help organizing the latent space and reducing the intrinsic dimension. Thus, SVD after retrieval has a better performance than the SVD-only approach.
>
> **Response to L1:**
>
> Thanks to the reviewer’s suggestions. During the rebuttal period, we found that using attention probability to select few patches is technically sound. Using 10% patches might saturate and the performance does not increase. This limitation is no longer severe based on the observation. We will test more datasets to see if this conclusion is consistent and will post the result as soon as it is available.

---

> > ### Comment · Reviewer_7vyh · 2023-08-13
> >
> > I have gone through the authors' rebuttal response. The authors have provided clarifications around the computational complexity of RAM-MIL in dynamic test set, intrinsic dimensions and the typo around having access to instance labels of target vs source set. It is interesting to see the gap between accumulated probabilities for 10% vs 20% of patches for Camelyon datasets (where signal is often in small foci) vs NSCLC subtyping (where signal is more diffused). I would have expected lower difference between 10% and 20% in Camelyon compared to NSCLC, however the results here show the opposite (2-1.5% in Camelyon vs 0.003% in NSCLC). Do the authors have a reason behind why this could be happening?

---

> > > ### Author Response · Authors · 2023-08-15
> > >
> > > Thanks for the responses!
> > >
> > > We agree that the positive WSIs in NSCLC have larger proportion of positive patches, as indicated by *Wang et al*. As for the attention mismatch problem, there are two major reasons:
> > > 1. In a trained attention-based multiple instance learning (ABMIL) model, a high attention probability does not necessarily represent a positive patch. This problem was discussed theoretically in Bayes-MIL (*Cui et al*). The high attention probability only converges to positive patches under strict assumptions or with a strong regularization of the model.
> > > 2. The logic of ABMIL is, when only one patch is positive, the slide is regarded as positive. Therefore, the generated attention values could possibly highly concentrate on a few patches, if a model is not well regularized.
> > >
> > > This explains why the accumulated attention values have no correlations with the size of positive area. They are influenced more by the model optimization process. However, this does not affect that we use the attention probability for calculating the OT, as negative patches also contain information and contribute to the similarity between slides.
> > >
> > > *Wang et al*. Targeting tumor heterogeneity: multiplex-detection-based multiple instance learning for whole slide image classification. Bioinformatics.
> > > *Cui et al*. Bayes-MIL: A New Probabilistic Perspective on Attention-based Multiple Instance Learning for Whole Slide Images. ICLR 2023.

---

> > > > ### Comment · Reviewer_7vyh · 2023-08-16
> > > >
> > > > Thank you for the clarification. After going through this response as well as the original one, I keep my existing rating.

---

### Official Review · Reviewer_q4PZ · 2023-07-04

**Soundness:** 3 good
**Presentation:** 3 good
**Contribution:** 3 good
**Rating:** 7
**Confidence:** 4

**Summary:**

The paper proposes a new Multiple instance learning method called RAM-MIL. The proposed method is a two-stage approach which mainly involves the following steps: 1. Train an existing MIL model on the D0 to extract feature representations of each instance 2. Retrieve nearest neighbor from retrieval set to form merged bag representation 3. Train another classifier on the merged bag representation.  The key component in the proposed method is the use of Optimal-Transport to measure bag-to-bag distance, retrieve nearest neighbor bag and form merged bag representation. The proposed approach improves out-of-domain performance while also maintaining good in-domain performance.

**Strengths:**

1. writing is mostly clear and easy to follow
2. related work section is comprehensive
3. the idea of using OT for measuring bag-to-bag distance and forming merged representations seems effective.

**Weaknesses:**

while the proposed method improve performance over the compared baselines (Table1), it also introduces additional steps on top of these baseline (finding nearest neighbors, forming merged bag representation and then train another bag classifier), in this sense, the additional performance gain is not that surprising.

**Questions:**

1. Did you describe how Figure 3 is obtained?
2. Can you clarify your experiment setting for the baseline methods (DSMIL, CLAM, TransMIL, Bayes-MIL) in Table1, did they touch the retrieval set at all?



**Limitations:**

the efficiency of the algorithm due to solving optimal transport problem is acknowledged.

---

> ### Author Rebuttal · Authors · 2023-08-10
>
> Thanks for the comments. Here are the responses to weaknesses (W), Questions (Q) and limitations (L).
>
> **Response to W1:**
>
> Thanks for the comments. As retrieval (also dubbed as external memory) has achieved great successes in applications like large language models [1][2], we believe it is worth to exploit its application in other important domains, like the challenging histopathological diagnosis. Our paper first provides the theoretical understanding, methodology, algorithms, and analysis for retrieval with multiple instance learning. For the algorithm part of this topic, it is necessary to know how retrieval could be done and how the retrieved knowledge could be merged into the original model.
>
> Note that we provide multiple choices to execute our retrieval algorithm including using L2, approximate-OT, Haursdorff distance and OT. Running the approximate version (L2 or approximate-OT) of the “additional steps” introduce negligible overhead in “finding nearest neighbor” (see stats below). Merging bag representation and train a linear bag classifier causes trivial computational overhead with modern consumer-grade GPUs. **Therefore, the additional gain does not introduce much overhead. Note that TransMIL and Bayes-MIL are heavier model with larger model size and longer running time.** We will update the parameter comparisons and running time comparisons later to show our advantage.
>
> || L2 | Hausdorff | Approximate-OT | Full OT |
> |-------------|---------|---------|---------|---------|
> | Time (s)        | 0.001  | 1.4  | 0.083| 0.498|
>
> Inspired by your comments, we think it is a good future work to study how to merge the retrieved data to MIL, in an online fashion (which takes even less time). We perform a preliminary test on this with retrieval integrated into the training pipeline of RAM-MIL. Here are the preliminary experimental results, which shows a promising online approximation to the offline-RAM-MIL:
>
> || C16-AUC | C16-ACC | C17-AUC | C17-ACC |
> |-------------|---------|---------|---------|---------|
> | RAM-MIL-Online        | 0.9263  | 0.875  | 0.7686 | 0.7453 |
> | RAM-MIL       | 0.9451|0.9200|0.7974|0.7795|
>
> [1] Borgeaud et al. Improving language models by retrieving from trillions of tokens. ICML 2022.
> [2] Gun et al. REALM: Retrieval-Augmented Language Model Pre-Training. ICML 2020.
>
> **Response to Q1:**
>
> To compute the intrinsic dimension, we obtain the slide features of each method then feed to the estimator provided by [Facco] for computation.
> Here are the procedures for obtaining slide features for different methods:
> CLAM: we take the weighted patch features as the slide feature.
> Mixup: we randomly pick slide features at the same class as a neighbour, then merge the two features with convex combination.
> The rest: these variants merge the slide features of source and retrieval, with retrieval implemented by different methods.
>
> Facco, et al. Estimating the intrinsic dimension of datasets by a minimal neighborhood information. NeurIPS 2017.
>
> **Response to Q2:**
> There is no setup of “retrieval set” in the baseline method, as this paper is the first to consider retrieval in multiple instance learning. However, **the baseline methods and our method access the same data under the in-domain setting during training.** The retrieval set is only taken from the in-domain dataset (e.g., training set and validation set of Camelyon16).
>
> For out-of-domain setting, there is a retrieval dataset providing unlabeled out-of-domain data for retrieval. For a fair comparison, we add a method [Yang et al] that also utilize the retrieval dataset and adversarial training in the experiments with Camelyon16 and Camelyon17, as suggested by reviewer 5E1m. Here are the results:
>
> || AUC | ACC |
> |-------------|---------|---------|
> | Yang et al        | 0.6786 | 0.5867  |
> | RAM-MIL        | **0.7974**  | **0.7795** |
>
> Yang et al. Double adversarial domain adaptation for whole-slide-imageclassification. MIDL 2021.
>
> **Response to L1**
>
> We add the efficiency results. Please refer to "response to W1".

---

> > ### Comment · Reviewer_q4PZ · 2023-08-14
> >
> > I have read the authors' response. The response addresses my concerns regarding the additional overhead. I also appreciate the elaboration on the motivation of introducing the retrieval procedures.
> >
> > Q: how did you get the RAM-MIL-Online results? Will that be describe in the paper?
> >
> > I will consider raising my score to 7.

---

### Official Review · Reviewer_yGH7 · 2023-07-05

**Soundness:** 3 good
**Presentation:** 3 good
**Contribution:** 3 good
**Rating:** 6
**Confidence:** 4

**Summary:**

The authors propose a novel MIL framework which integrates Wasserstein Distance and Optimal Transport to match instance representations across bags. This method can be further leveraged to match representations across bags from different domains, thereby enabling Cross Domain Adaption.



**Strengths:**

The authors' RAM-MIL method achieved state-of-the-art performance in terms AUC and accuracy on CAMELYON16 and CAMELYON17, as well as MUSH1, MUSK2, FOX, and ELEPHANT. The authors provide interesting insights on the dimension reduction properties of their method. Finally, the authors demonstrate ways to interpret and visualize their method.

**Weaknesses:**

1. The authors might have unmasked themselves in the double-blind review process by using the same pathology image in Figure 1 as Bayes-MIL

Minor Problems:
1. The authors state Theorem 2, without clearly stating the proof is the supplement. The authors seem to state the proof of a similar result for the feature space is in the supplement
2. Minor typos such as "which which" (line 308)


**Questions:**

1. The authors state that "the calculation of OT quite time-consuming", which is why they pre-selected the top20% of instances. Can the authors give training time comparisons between OT, l2, Hausdorff and Approx-OT? This will help readers understand the practical tradeoffs between l2 and OT.
2. In the supplement, the authors provided a table of results for OT with top10% and top20% of instances. The performance differences seem minor. What would happen with top30% of instances or is this a problem-specific choice depending on the type of disease and relative coverage of disease features in the WSI images?
3. I am not sure if I follow the authors' reasoning in section 5.2 and the corresponding Table 3. The authors state that the intrinsic dimension of CLAM is 5-7 on CAM16 and CAM17, then why would they cap merged bag representation to 5d vector? Would this not impact CLAM more than RAM-MIL and affect its performance? What if it was capped to 7d?


**Limitations:**

The authors could discuss the general applicability of RAM-MIL. Is it only to close problems such as CAM16 and CAM17? Can it help with WSI and bag labels that are different diseases?

---

> ### Author Rebuttal · Authors · 2023-08-10
>
> Thanks for the comments. Here are the responses to weaknesses (W) and Questions (Q).
>
> **Response to W1:**
>
> Thanks for mentioning the potential issue. Please note that we follow the double-blind reviewing rules and the submission does not contain any identifying information. For Figure 1, some slides are borrowed from visualization of Bayes-MIL library, which are based on OpenSlide and CLAM library. As Bayes-MIL is a published work and the codes are accessible online (https://github.com/ralphc1212/bayes-mil), we believe it does not violate the anonymous rule to use the same visualization tool in our submission. We will cite Bayes-MIL in the manuscript in Figure 1.
>
> **Response to W-Minor 1:**
>
> As indicated in Line 480-481 of Page 13, supplemental, to get the same results for features, we only need to assume a probability measure over the instance representations h and replacing all X in the proof with H. We will provide a theorem 2 and a formal proof in the supplemental.
>
> **Response to Q1:**
>
> Thanks for the advices. Here are the time comparisons for calculation one pair of slides under different metrics:
>
> || L2 | Hausdorff | Approximate-OT | Full OT |
> |-------------|---------|---------|---------|---------|
> | Time (s)        | 0.001  | 1.4  | 0.083| 0.498|
>
> Note that Approximate-OT provide a decent trade-off between computation time and performance. Next we show using a few patches (like 10%), the performance might saturate due to the most important information is stored in a few patches. Therefore, it might be a good choice to use a few patches for OT computation.
>
> **Response to Q2:**
>
> Thanks for the advices. We calculate the cumulative attention probability value for 10% and 20% of patches:
>
> || C16 | C17 | TCGA-NSCLC |
> |-------------|---------|---------|---------|
> | 10% of patches        | 0.9661  | 0.9667  | 0.9969|
> | 20% of patches    | 0.9882  | 0.9805 | 0.9999|
>
> This indicates most useful information are already in the 10% data, which explains why the performance does not change much when increasing the data amount.
>
> **Response to Q3:**
>
> Section 5.2 is to show directly applying simple dimensionality reduction method like SVD is not useful in improving the performance. By using retrieval then SVD, the performance is significantly better. We add an experiment with capped dimensionality 7:
>
> || C16-AUC | C16-ACC | C17-AUC | C17-ACC |
> |-------------|---------|---------|---------|---------|
> | CLAM        | 0.8821  | 0.6688  | 0.7568  | 0.5474  |
> | RAM-MIL    | **0.9312**  | **0.905** | **0.7641** | **0.7434** |
>
> **Response to limitations:**
> We also add experiments on other datasets: in-domain classification on TCGA-NSCLC, in-domain tumor stage classification with Camelyon17, in-domain classification on CPTAC-LSCC, domain adaptation from CPTAC-LSCC to CPTAC-UCEC.
>
> Here are the results:
>
> TCGA-NSCLC (in-domain):
>
> || AUC |ACC |
> |-------------|---------|---------|
> | CLAM        | 0.9420  | 0.8640  |
> | Scaling ViT | 0.9516  | 0.8821  |
> | TransMIL    | **0.9603**  | 0.8835  |
> | Bayes-MIL   | 0.9451  | 0.8965  |
> | RAM-MIL     | 0.9497  | **0.8988**  |
>
> Camelyon17 tumor stage classification (in-domain):
> || AUC |ACC |
> |-------------|---------|---------|
> | CLAM        | 0.7803  | 0.60  |
> | Bayes-MIL | 0.8070 | 0.64  |
> | RAM-MIL    | **0.8132**  | **0.65**  |
>
> LSCC as in-domain dataset and UCEC as out of domain dataset:
> || LSCC-AUC | LSCC-ACC | UCEC-AUC | UCEC-ACC |
> |-------------|---------|---------|---------|---------|
> | CLAM       | 0.9500 | 0.9285 | 0.4986  | 0.6520  |
> | RAM-MIL | **0.9667** | **0.9382** | **0.6056** | **0.6527**  |

---

> > ### Comment · Reviewer_yGH7 · 2023-08-16
> >
> > I thank the authors for answering my questions and conducting experiments on more datasets and ablation studies. These results confirm my previous opinion and my rating is unchanged.

---

### Official Review · Reviewer_WXnS · 2023-07-07

**Soundness:** 2 fair
**Presentation:** 3 good
**Contribution:** 3 good
**Rating:** 4
**Confidence:** 4

**Summary:**

This work presents a method for multiple instance learning-based method for slide retrieval based on optimal transport, called RAM-MIL. The idea is as follows: 1) first train a standard ABMIL model for supervised classification, 2) pre-extract the features from ABMIL to get slide-level features, 3) compute the OT map via SInkhorn's between bag of instance representations (using the attention weights to calculate the OT) within the retrieval set, 4) merge representations followed by training a logistic classifier. RAM-MIL was trained and evaluated in CAMELYON16 and CAMELYON17, with ablation experiments conducted with other OT-based retrieval methods.



**Strengths:**

- This work represents a new method for augmenting MIL with optimal transport, particularly for handling out-of-distribution data. The idea of computing optimal transport maps between instances and bags in histopathology have successfully been performed in prior literature such as HHOT (Yeaton et al. 2022 MIDL), as well as with regard to tackling domain adaptation problems such as Falahkheirkhah et al. 2023 MIDL, Domain adaptation using optimal transport for invariant learning using histopathology datasets. Distinct from prior works is the further application of OT with feature merging at the bag-level for evaluating out-of-domain performance in slide-level classification. At the same time, this work still brings several interesting ideas on how to leverage the attention weights as the probability measures for computing OT maps.
- Beyond the main paper, the supplement is also nicely organized and includes additional ablation experiments regarding % of attention weights used and different merge functions. Further evaluation of patch-level localization performance was also performed, which is appreciated in baselines such as CAMELYON16 (commonly used for slide-level performance only).
- Figures are illustrative and well-designed.

**Weaknesses:**

1. Regarding novel contribution #1: On line 67, one of the contributions state that "this work is among the first to investigate the out-of-domain performance for MIL, which is vital issue for the application of MIL in automated medical diagnosis with WSI. Our empirical study exposes the risk of recent MIL algorithms under distribution shifts." This claim is not substantiated and untrue, as most clinical studies in computational pathology do consider the problem of out-of-domain performance of MIL models (for instance, Campanella et al. 2018 Nature Med, Lu et al. 2018 Nature BME, Courtiol et al. 2019 Nat Med., etc). Regarding out-of-domain performance for image retrieval, works such as Yottixel (Kalra et al. 2020, Medical Image Analysis) and SISH (Chen et al. 2022, Nature BME) have also performed quantitative evaluation on out-of-domain external cohorts.
2. Baselines: Though methods such as CLAM and Bayes-MIL were used as comparisons, this work lacks comparisons with other important baselines for slide retrieval in histopathology such as SISH and Yottixel, and OT-based pathology works such as HHOT (most similar to this work conceptually). Other missing baselines include performance of conventional ABMIL (that was used to train RAM-MIL), and also linear probe analysis on top of the already pre-extracted ABMIL features. The reviewer would also appreciate hyper-parameters used in linear probe analysis of this work, which may also importantly influence results.
3. Retrieval: Though the contributions of this work are posed as solving a retrieval problem, the evaluation (task, metrics) is focused on solving a popular but narrow task in histopathology (CAMELYON16 - a needle-in-a-haystack problems). Only one dataset is used for evaluation of this method, which may not be representative of broader slide-level retrieval tasks in histopathology. Lastly, one potential practical limitation is the computational complexity of using OT for retrieval, which should be commented on in the rebuttal.

I am overall positive about this method, as this work brings an interesting idea regarding augmenting OT with attention weights from ABMIL in histopathology, my main concern of this work is the lack of evaluation on other datasets such as TCGA, CPTAC, NLST, GTEX, etc. - which can be readily used for evaluating out-of-domain performance and slide-retrieval in a pan-cancer setting, as well as missing baselines.

**Questions:**

1. How well does this method perform against other baselines such as SISH / Yottixel for slide retrieval? ABMIL with and without linear probe?
2. There exist many histopathology datasets that can be combined to form external cohorts. The findings of this work would be strengthened if more datasets and cancer types were evaluated.

**Limitations:**

Limitations of this work were addressed.

---

> ### Author Rebuttal · Authors · 2023-08-10
>
> Thanks for the comments. Here are the responses to weaknesses (W) and Questions (Q).
>
> **Response to W1:**
>
> Thanks for mentioning the related works. We will add them into the related works sections.
>
> For the claim of contribution, we meant to say, “the out-of-domain performance of Attention-based Multiple Instance Learning (ABMIL)” rather than the first work considering out-of-domain problem for whole slide image. As it might sound ambiguous, we will change the claim of contribution to a more precise one.
>
> **Response to W2 and Q1:**
>
> For the information retrieval works (Yottixel and SISH) of whole slide image, we will add them into the related works section. However, these two methods are not directly comparable, as they focus on the retrieval problem while our goal is to use retrieval to enhance the weakly supervised classification (MIL). An executable way is to replace the OT-retrieval module in RAM-MIL with these retrieval methods. As there is no available implementation of Yottixel, and SISH is a more advanced retrieval method, we integrate SISH retrieval into the RAM-MIL process. The experiment runs on Camelyon17 (including 5 hospitals), with 2 hospitals as the in-domain dataset and the rest being the out-of-domain dataset. Here are the experimental results:
> ||AUC| ACC|
> |-----------|--------|--------|
> |CLAM|0.8155|0.7222|
> |CLAM-SISH|0.7647|0.7444|
> |RAM-MIL|**0.8209**|**0.7667**|
>
> For the OT-based pathology work, we add HHOT into the related works section. HHOT only considers comparing whole slide images with uniform weight-based OT. As they mentioned kNN as a discriminative method, we compare HHOT-kNN with RAM-MIL (C16 as in-domain, C17 as out of domain):
> ||C16-AUC|C16-ACC|OOD-C17-AUC|OOD-C17-ACC|
> |----------------|---------|---------|---------|---------|
> |HHOT-kNN (k=1)|0.7007|0.7318|0.6939|0.7173|
> |HHOT-kNN (k=3)|0.7618|0.7544|0.7263|0.7523|
> |ABMIL|0.9010|0.875|0.7287|0.7189|
> |ABMIL w/ linear probe|0.9107|0.8725|0.7312|0.7077|
> |RAM-MIL|**0.9451**|**0.9200**|**0.7974**|**0.7795**|
>
> We also include ABMIL with/without linear probe in the above table.
>
> **Response to W3 and Q2:**
>
> We add experiments with in-domain classification on TCGA-NSCLC, in-domain tumor stage classification with Camelyon17, in-domain classification on CPTAC-LSCC, domain adaptation from CPTAC-UCEC to CPTAC-LSCC.
> Here are the results:
>
> TCGA-NSCLC (in-domain):
>
> || AUC |ACC |
> |-------------|---------|---------|
> | CLAM        | 0.9420  | 0.8640  |
> | Scaling ViT | 0.9516  | 0.8821  |
> | TransMIL    | **0.9603**  | 0.8835  |
> | Bayes-MIL   | 0.9451  | 0.8965  |
> | RAM-MIL     | 0.9497  | **0.8988**  |
>
> Camelyon17 tumor stage classification (in-domain):
> || AUC |ACC |
> |-------------|---------|---------|
> | CLAM        | 0.7803  | 0.60  |
> | Bayes-MIL | 0.8070 | 0.64  |
> | RAM-MIL    | **0.8132**  | **0.65**  |
>
> LSCC as in-domain dataset and UCEC as out of domain dataset:
> || LSCC-AUC | LSCC-ACC | UCEC-AUC | UCEC-ACC |
> |-------------|---------|---------|---------|---------|
> | CLAM       | 0.9500 | 0.9285 | 0.4986  | 0.6520  |
> | RAM-MIL | **0.9667** | **0.9382** | **0.6056** | **0.6527**  |

---

> > ### Author Response · Authors · 2023-08-17
> >
> > We also updated the related works including the mentioned methods by the reviewer, drafted as “Update of related works” in responses to reviewer 5E1m. Please check.
> >
> > -Authors

---

### Official Review · Reviewer_5E1m · 2023-07-19

**Soundness:** 3 good
**Presentation:** 3 good
**Contribution:** 2 fair
**Rating:** 5
**Confidence:** 5

**Summary:**

This paper introduces the Retrieval-AugMented MIL (RAM-MIL) framework, which addresses the challenge of performance deterioration in MIL models when tested on an out-of-domain test set. The proposed framework achieves state-of-the-art performance in both in-domain and out-of-domain scenarios. The RAM-MIL framework integrates Optimal Transport as the distance metric for nearest neighbor retrieval and uses the transportation matrix derived from OT to render the retrieval results interpretable at the instance level. The paper also proves a theorem that demonstrates the reduced input dimensions lead to improved MIL performance and proposes the RAM-MIL algorithm to learn a low intrinsic dimension feature space and enhance the model generalization, especially for out-of-domain data. The proposed methodology is evaluated on whole slide image (WSI) datasets and general MIL datasets.

**Strengths:**

This article relates the input data dimension to the performance of MIL and provides a novel theoretical explanation. This perspective is very innovative.

The article addresses the unsupervised domain adaptation problem by retrieving instances with similarities from the unlabeled domain and merging them with instances from the labeled domain. This approach impresses me deeply.

**Weaknesses:**

1.The author mentioned that they are the first to explore the problem of domain adaptation in pathological image domains. However, there have been some existing works in this area, including domain adaptation for whole slide images. For example:

[1] Collaborative Unsupervised Domain Adaptation for Medical Image Diagnosis

[2] Double adversarial domain adaptation for whole-slide-image classification

[3] Adversarial Domain Adaptation for Classification of Prostate Histopathology Whole-Slide Images.

[4] Unsupervised Domain Adaptation for Classification of Histopathology Whole-Slide Images

2.(1) As mentioned in Comment 1, there have been numerous studies in this field. However, this article lacks a detailed review of domain adaptation issues in pathological images, including whole slide images. It also fails to provide a comprehensive explanation of the differences between the proposed method in this article and those previous studies.

(2) Additionally, the experiments in this article only compare against baseline and self-variants, without comparing with the currently most advanced methods available.

3.The author mentioned, "Our theoretical result based on Wasserstein distance in the input space shows a negative correlation between **MIL performance** and the **input data dimension**, which motivates us to propose the novel RAM-MIL framework based on the OT."

(1) What does the term "input data dimension" refer to? Is it the feature dimension of the input instances? Does "MIL performance" refer to the performance of unsupervised domain adaptation or the performance of MIL itself?

(2) As far as I know, most current MIL methods first use a pretrained self-supervised network to map instances from the image domain to the feature domain, and the dimension of these features can be defined by the pretrained self-supervised network. Therefore, can't the input data dimension issue raised in this article be simply solved by modifying and defining it through a pretrained network?

(3) In conclusion, I haven't understood the relationship between the "input data dimension" and "MIL performance" in this article. Why does reducing the input data dimension improve performance? To what extent should it be reduced at least? How does this article's method determine the optimal input data dimension?

4.Is the Merge operation performed by weighted averaging? How is the Merge operation specifically completed?

I'm looking forward to the author's reply, and after my questions are answered, I will reconsider my rating.

**Questions:**

Please refer to Weaknesses.

**Limitations:**

Please refer to Weaknesses. I'm looking forward to the author's reply, and after my questions are answered, I will reconsider my rating.

---

> ### Author Rebuttal · Authors · 2023-08-10
>
> Thanks for the comments. Here are the responses to weaknesses (W).
>
> **Response to W1 and W2.1:**
>
> Thanks for mentioning the related works. We will add discussions about [1, 2, 3, 4] into the related works and [2] will be added to comparisons in the evaluation.
>
> Note that [1, 2, 3, 4] are **NOT** attention-based multiple Instance Learning model for whole slide image classification. [1, 3, 4] are domain adaptation methods for the instance (patch)-level classifier rather than multiple instance learning model. The instance-level classifiers neglect the global correlation between patches and require instance-level labels for training the classifier. The most related work is [2], where patch features of a whole slide image are aggregated using the Fisher vector encoding and global pooling. The fundamental difference between [2] (also noted as Yang et al in other responses) and ours is that, our method is based on attention-based multiple instance learning where the patch-level features are aggragated by learnable attentions.
>
> Nevertheless, we add an experiment for comparing with [2] which will be added to the manuscript. As the datasets (CCI dataset and Warwick HER2 challenge) used in [2] are not accessible now, we evaluate [2] on with our experimental setting, using Camelyon16 as the in-domain dataset and Camelyon17 as the out of domain dataset. Accuracy and AUC of [2] are **58.67** and **67.86**, outperformed by RAM-MIL (Acc: **77.95**, AUC: **79.74**).
>
> **Response to W2.2**
> > Additionally, the experiments in this article only compare against baseline and self-variants, without comparing with the currently most advanced methods available.
>
> We did compare with most advanced methods including CLAM, TransMIL and BayesMIL. In the rebuttal, we also add experiments comparing with [2] (domain adaptation for MIL, see responses to W1 and W2.1), HHOT (OT-method for WSI, see responses to reviewer WXnS) and SISH (Nearest Neighbour Search method, see responses to reviewer WXnS).
>
> **Response to W3.1**
>
> Thanks for spotting one critic motivation of our work. The sentence at the line 70, page 2 is not accurate and will be clarified in the updated version. “input data dimension” should be “the *intrinsic dimension* of input data”, which should be distinguished from the normal dimension. We assume that the data reside on a low-dimensional manifold in the input data space, so the intrinsic dimension is less than the normal dimension.  As mentioned in the paper (line 111-113, page 3), the intrinsic dimension is measured by how many dimensions are needed to represent the data without much loss of information [Facco, et al.]. The MIL performance refers to the approximation error of an MIL model.
>
> Facco, et al. Estimating the intrinsic dimension of datasets by a minimal neighborhood information. NeurIPS 2017.
>
> **Response to W3.2**
>
> Thanks for the advices. Directly reducing the normal dimension to the estimated intrinsic dimension is a brute-force method to improve the performance. To validate if this argument holds, we use a masked auto-encoder (MAE) with output dimension to be 5 (the estimated intrinsic dimension) to extract the features from patches. To ensure a good quality of extracted features, we fine-tune the MAE with all patches from Camelyon16 for 60 epochs. The performance of CLAM fitted on the extracted features is at the second row of the Table:
>
> | Model       | C16-AUC | C16-ACC | OOD-C17-AUC | OOD-C17-ACC |
> |-------------|---------|---------|-------------|-------------|
> | CLAM-SVD    | 0.9216  | 0.6625  | 0.7486      | 0.4991      |
> | CLAM-MAE    | 0.8285  | 0.7368  | 0.7325      | 0.6200      |
> | RAM-MIL-SVD | **0.9273**  | **0.8475**  | **0.7509**      | **0.7053**      |
>
> Simply reducing the output dimension to the estimated intrinsic dimension cannot provide a decent performance, which provides a justification for our RAM-MIL method.
>
> **Response to W3.3**
>
> *The theoretical results show that a lower intrinsic dimension would lead to a better performance of the multiple instance learning model. This statement is intuitively correct.* We list the intuitions as follows:
>
> 1) Lower intrinsic dimension may mean that the data lies on a simpler manifold or subspace. As the complex of the subspace is lower, learning is easier.
>
> 2) Lower intrinsic dimension might also imply that the data is less noisy and redundant, which could make learning easier.
>
> Although there exist some methods to test the intrinsic dimension of data, there is no principled method to reduce the intrinsic dimension and determine the optimal intrinsic dimension. Our paper proposes a retrieval-augmented MIL that could potentially reduce the intrinsic dimension and it is validated in the paper (Figure 3 and Table 3).
>
> **Response to W4**
>
> As indicated in line 155, page 4, the merging function is a convex combination. We have tested different configurations of the merging function, shown in Table 6, page 14, supplemental materials. We will make this clear in the updated version.

---

> > ### Comment · Reviewer_5E1m · 2023-08-15
> >
> > I appreciate the author's thoughtful response, and I have carefully reviewed the author's rebuttal. Overall, I partially acknowledge the novelty of the Retrieval-Augmented MIL method proposed in this paper and the research perspective that 'lower intrinsic dimension would lead to a better performance of the MIL model.' As I mentioned in my review, I have two main concerns about this article.
> >
> > **The first point** is the need for a more detailed overview and comparison of domain adaptation in WSI Classification. Domain adaptation in pathological image domains has been developed for a considerable amount of time, with various technical methods at both slide-level and instance-level [1], such as solutions involving stain normalization, feature-level domain adversarial learning, spatial-based cycle migration, and others. This paper introduces a Retrieval-Augmented solution at the feature level, and I acknowledge the novelty of this approach. However, the authors did not comprehensively and systematically review and compare the literature in this domain in their paper, while highlighting the innovation and advantages of their method. In the experimental comparison, it is important to step beyond the 'self-variant,' including a comparison with state-of-the-art methods from other paradigms within this domain. For instance, domain adaptation in WSI Classification typically focuses on addressing staining variations. How does Retrieval-Augmented at the feature level compare in terms of advantages? How does the performance of feature-level domain adversarial learning perform under the same experimental settings as this paper? In summary, I believe that the current review and comparison of other paradigms within the same domain are not comprehensive enough in this paper.
> >
> > **The second point** is that the study of feature dimensions in this paper is not sufficiently thorough and comprehensive. As I mentioned in my review,
> > > most current MIL methods first use a pretrained self-supervised network to map instances from the image domain to the feature domain, and the dimension of these features can be defined by the pretrained self-supervised network. Therefore, can't the input data dimension issue raised in this article be simply solved by modifying and defining it through a pretrained network?
> >
> > Although the author validated this idea by constructing a simple MAE in the rebuttal, this is not comprehensive enough. Currently, most MIL articles on the Camelyon 16 dataset, such as [2,3,4,5], use pretrained self-supervised models or ImageNet pretrained models to extract features beforehand and achieve good performance (much higher than the C16-AUC 0.8285 mentioned in the author's rebuttal). As far as I know, the dimensions of these self-supervised or ImageNet pretrained model features can be customized through simple network modifications during training. The claim that reducing feature dimensions can enhance MIL's performance needs more comprehensive validation. Moreover, under the same feature dimensions, how advantageous is the Retrieval-Augmented method?
> >
> > Another issue is determining the optimal feature dimension. In the rebuttal, the author conducted an experiment using MAE with a feature dimension of only 5 (the estimated intrinsic dimension). How is this 'estimated intrinsic dimension' calculated? Generally, self-supervised feature dimensions are around 128-256 dimensions. The author mentions that 'there is no principled method to reduce the intrinsic dimension and determine the optimal intrinsic dimension. Our paper proposes a retrieval-augmented MIL that could potentially reduce the intrinsic dimension and it is validated in the paper.' This is quite confusing.
> >
> > Finally, I would like to express my gratitude once again for the author's efforts and thoughtful response. I am very interested in this article and am willing to consult and discuss with the author, reviewers, and the AC. I look forward to the author's more comprehensive summary and response, as well as the sharing and guidance from other reviewers and the AC. According to the standards of top academic conference, I anticipate that this article will become a leading piece of research in the field.
> >
> > [1] Srinidhi, Chetan L., Ozan Ciga, and Anne L. Martel. "Deep neural network models for computational histopathology: A survey."  Med Image Anal. 67 (2021): 101813.
> >
> > [2] Li, Bin, Yin Li, and Kevin W. Eliceiri. "Dual-stream multiple instance learning network for whole slide image classification with self-supervised contrastive learning." CVPR. 2021.
> >
> > [3] Shao, Zhuchen, et al. "Transmil: Transformer based correlated multiple instance learning for whole slide image classification." NeurIPS 34 (2021): 2136-2147.
> >
> > [4] Chen, Richard J., et al. "Scaling vision transformers to gigapixel images via hierarchical self-supervised learning." CVPR. 2022.

---

> > > ### Author Response · Authors · 2023-08-16
> > > **Responses to the first point**
> > >
> > > Many thanks to the reviewer for acknowledging the novelty of our approach. As claimed in our response, we agree with the reviewer that we need to add more discussions in the updated version about existing works in domain adaptation for whole slide images including but not limited to [1,2,3,4].  However, as we claimed in the response, and acknowledged in the survey paper the reviewer mentioned (Srinidhi et. al. – *domain adaptation and stain variability methods, Table 5*), a vast majority of existing methods are based on fully supervised learning on patches, instead of weakly-supervised learning on slides as in our paper. Specifically, fully supervised learning requires patch-level labels, which are usually not accessible in real world. Thus, those papers are not directly comparable with our method which targets at the multiple instance learning setting for slide-level classification, indicated by our submission title. Here are our responses to the two questions.
> > >
> > > >For instance, domain adaptation in WSI Classification typically focuses on addressing staining variations. How does Retrieval-Augmented at the feature level compare in terms of advantages?
> > >
> > > **Staining transfer:** [Cho et al] and [Shaban et al] are two of the most popular stain transfer methods in pathological image domains, both of which focus on patch-level performance and fail to consider the whole-slide property of the pathological images. Regarding the comparison between methodologies, [Cho et al] and [Shaban et al] need the training of a generative adversarial network (with patches), a task renowned for its inherent difficulty in achieving stable training. In contrast, our proposed RAM-MIL approach relies upon an OT retriever, obviating the requirement for further training, and is more convenient to use. As [Cho et al] has no available code, we will adapt [Shaban et al] with least modification to fit the slide setting. The experimental comparisons with [Shaban et al] will be updated later. *[Updated at the bottom, 2023-Aug-17]*
> > >
> > > >How does the performance of feature-level domain adversarial learning perform under the same experimental settings as this paper?
> > >
> > > **Domain adversarial learning:** We have provided the experimental comparison between [Yang et al], a domain adaptation method based on adversarial learning mentioned by the reviewer before, and our RAM-MIL: “**We evaluated [Yang et al] on with our experimental setting, using Camelyon16 as the in-domain dataset and Camelyon17 as the out of domain dataset. Accuracy and AUC of [Yang et al] are 58.67 and 67.86, outperformed by RAM-MIL (Acc: 77.95, AUC: 79.74).**” Please see "response to W1 and W2.1".
> > >
> > > We sincerely appreciate the valuable comments of the reviewer, which have significantly enriched the depth and scope of our discussion pertaining to domain adaptation. We will add the above discussion to our paper to elucidate the distinctions between RAM-MIL and pre-existing methods with more clarity. We express our appreciation again for the positive appraisal of the novelty of our paper and we are willing to address any further concerns.
> > >
> > > *[Cho et al] Cho, Hyungjoo, et al. "Neural stain-style transfer learning using GAN for histopathological images." arXiv preprint arXiv:1710.08543 (2017).*
> > >
> > > *[Shaban et al] Shaban, M. Tarek, et al. "Staingan: Stain style transfer for digital histological images." 2019 Ieee 16th international symposium on biomedical imaging (Isbi 2019). IEEE, 2019.*
> > >
> > > *[Yang et al] Yang, Yuchen, et al. "Double adversarial domain adaptation for whole-slide-image classification." Medical Imaging with Deep Learning. 2021.*
> > >
> > > ---
> > > *[Update: 2023-Aug-17]* We run [*Shaban et al*] - a representative CycleGAN based staining transfer method in patch-level in the Camelyon17 domain adaptation setting. For obtaining the WSI features, we use averaged pooling. For fairness, the same classifier as RAM-MIL is used for [*Shaban et al*]. Here are the results:
> > > |-|AUC|ACC|
> > > |-|-|-|
> > > |CLAM-SISH| 0.7647| 0.7444|
> > > |[*Shaban et al*]| 0.6458| 0.6889|
> > > |RAM-MIL| **0.8209**| **0.7667**|
> > >
> > > RAM-MIL outperforms the patch-level staining transfer based method.
> > >
> > > Another drawback of patch-level staining transfer is, it takes unacceptable time for training a CycleGAN. Per-epoch training time is over 1 hour on a single A100 GPU with 80G memory with only about 5% patches (guided by the paper) from the WSI pool. We finish this experiment with multiple A100s running distributively. Increasing the percentage of patches would further increase the training cost.
> > >
> > > ---

---

> > > > ### Author Response · Authors · 2023-08-16
> > > > **Response to the second point**
> > > >
> > > > Our presentation might have led to a misunderstanding or confusion between the **intrinsic dimension** and the **normal dimension**. **The proposed RAM-MIL framework does not contain any operation of dimension reduction and does not need to determine the optimal dimensionality.** Here we would be glad to make the following clarifications:
> > > >
> > > > 1. The dimensionality that reviewer mentioned (modifying the dimension by adjusting feature extractor) is **normal dimension**.
> > > > 2. The **intrinsic dimension** is **NOT** **normal dimension**.
> > > > 3. Reducing the **intrinsic dimension** could improve the MIL performance. (justified by Theorem 1 and with intuition explained in “Response to W3.3”). But reducing **normal dimension** might not and it is not our claim. (The typo is corrected in “Response to W3.1”. Thanks to the reviewer.)
> > > > 4. Reducing **normal dimension** does not necessarily reduce the **intrinsic dimension**.
> > > >
> > > > The popular references of intrinsic dimension includes [Facco et al], [Aghajanyan et al] and [Pope et al].
> > > >
> > > > Here are the detailed explanations and examples:
> > > >
> > > > >The claim that reducing feature dimensions can enhance MIL's performance needs more comprehensive validation.
> > > >
> > > > That is not our claim. Our theoretical and empirical results about intrinsic dimension is to help us understand the proposed methodology, rather than draw a conclusion “reducing the normal dimension of input could leads to a better performance”. Because the intrinsic dimension and normal dimension are different things, as indicated in “Responses to W3.1”.
> > > >
> > > > Our theoretical results show that, if the intrinsic dimension is reduced, the approximation error of multiple instance learning is reduced. The intrinsic dimension of some vectors can directly be computed by methods like [Facco, et al.].
> > > >
> > > > **EXAMPLE:**
> > > >
> > > > Let us make an intuitive example for this theoretical result, suppose there is a bag (or a whole slide) of instances (or patches). Each instance is a 224x224 image cropped from the whole slide. We use some feature extractor (ResNet-50, MAE, etc.) to obtain a 1024-dimensional feature for each instance. Normally, this bag of features is fed into an MIL model for classification.
> > > >
> > > > Suppose the intrinsic dimension of the 1024-dimensional features is 20. Our theoretical results show that, if we could reduce this value, the performance of MIL model will be better.
> > > >
> > > > Normal dimension: 1024.
> > > >
> > > > Intrinsic dimension: 20. <- *We want to reduce this dimension.*
> > > >
> > > > Then, let’s understand this argument with the following questions and answers, which are highly related to reviewer's questions:
> > > >
> > > > **1) How did we compute the intrinsic dimension?**
> > > >
> > > > Directly feeding the features to the estimator like [Facco, et al.] could get the result.
> > > >
> > > > **2) How do we reduce the intrinsic dimension?**
> > > >
> > > > There is no principled method. Manifold Mixup [Verma et al] performed on the features could help (see Figure 3). And our proposed retrieval-based method is proved to be better than Manifold Mixup.
> > > >
> > > > **3) Is the dimensionality reduction really performed?**
> > > >
> > > > No. We still feed the bag of 1024-dimensional features to the MIL model. No dimensionality reduction on the features is performed.
> > > >
> > > > **4) Why don’t we directly reduce the normal dimension by changing the output dimension of a feature extractor, or using SVD?**
> > > >
> > > > The rest 1004 (1024-20) dimensions still contain information that is useful for classification. Directly reducing the dimension could lose the information. The results in “Response to W3.2” verify this point.
> > > >
> > > > **5) Why do we still perform SVD in the experiment (Section 3.2)?**
> > > >
> > > > It’s to show that directly reducing the normal dimension to the estimated intrinsic dimension using methods like SVD does not work.
> > > >
> > > > **6) How is the estimated intrinsic dimension calculated? Why it is 5?**
> > > >
> > > > Our experiments in Section 3.2 show that our method could reduce the intrinsic dimension of Camelyon16 features to 5 to 7. Again, this value is computed by using [Facco, et al] on the merged features.
> > > >
> > > > We thank the reviewer for having an in-depth thinking and bringing active discussion, which helps us to understand our work better. We are willing to address any further concerns, if there are.
> > > >
> > > > *[Facco et al] Estimating the intrinsic dimension of datasets by a minimal neighborhood information. Scientific reports 2017.*
> > > >
> > > > *[Verma et al] Manifold mixup: Better representations by interpolating hidden states. ICML 2019.*
> > > >
> > > > *[Aghajanyan et al] Intrinsic Dimensionality Explains the Effectiveness of Language Model Fine-Tuning. ACL 2021.*
> > > >
> > > > *[Pope et al] The Intrinsic Dimension of Images and Its Impact on Learning. ICLR 2021.*

---

> > > > > ### Author Response · Authors · 2023-08-17
> > > > > **Update of related works**
> > > > >
> > > > > **Update of Related Works**
> > > > >
> > > > > We use abbreviation like [Name, Journal/Conference Year] here for saving space. Complete citations will be updated in the manuscript.
> > > > >
> > > > > **1. Domain adaptation**
> > > > >
> > > > > **1.1 Patch-level:** there have been a large volume of research studying the domain adaptation in patch-level, categorized as staining transfer and domain adversarial learning.
> > > > >
> > > > > **1.1.1 Patch-level staining transfer:** [*Cho et al, Arxiv 2017*], [*Shaban et al, ISBI 2019*] and [*Zanjani et al, MIDL 2018*] use generative adversarial networks (GAN) to learn the staining difference implicitly. Images are transferred to the target domain, by the trained generators in GAN. A simple neural network is then fit on the generated images for discriminative tasks.
> > > > >
> > > > > **1.1.2 Patch-level domain adversarial learning:** [*Lafarge et al. DLMIA, 2017*] and [*Ciga et al. Arxiv 2019*] apply gradient reversal method for adapting the model for discriminative tasks in medical images. [Ren et al] uses Siamese network in domain adversarial learning. [*Brieu et al, Arxiv 2019*], [*Kapil et al, MICCAI 2019*] and [*Gadermayr et al, TMI*] for transfer images between domains for segmentation tasks.
> > > > > These methods are not directly comparable with our methods, as they require patch-level labels during training, which is not realistic in real-world scenario. In experiment, we adapt [*Shaban et al*] with least modification, to compare with the retrieval based multiple instance learning.
> > > > >
> > > > > **1.2 Slide-level methods:**
> > > > > [*Yang et al*] propose a slide-level domain adaptation method that involves local and global adversarial loss to train a weakly supervised learning model. RAM-MIL outperforms this method in the evaluation.
> > > > >
> > > > > **1.3 Other methods:**
> > > > > Previous methods in slide-level classification (*Campanella et al. 2018 Nature Med*) and methods in other fields (*Lu et al. 2018 Nature BME, Courtiol et al. 2019 Nat Med.*) also consider the out-of-domain testing performance. However, they did not provide a method for domain adaptation, while RAM-MIL has. We compare RAM-MIL with several most advanced baselines in slide-level classification in our experiments.
> > > > >
> > > > > **2. Retrieval methods:**
> > > > >
> > > > > Recent works also study how to efficiently retrieve relative WSI from database. Yottixel (*Kalra et al., MIA 2020*) builds a search engine for indexing WSIs at scale. SISH (*Chen et al, Nature BME 2022*) uses a tree structure for fast search of WSI followed by an uncertainty-based ranking algorithm for retrieval. In experiment, we integrate the open-sourced SISH as a search engine into our RAM-MIL pipeline for comparisons.
> > > > >
> > > > > HHOT (*Yeaton et al. MIDL 2022*) proposes to use optimal transport (OT) as a distance measure to compare different WSIs, or different WSI datasets. Our paper, focusing on classification tasks, studies the principle why OT is suitable for WSI classification and propose a retrieval-based classification process. We compare RAM-MIL and HHOT in our experiments.

---

> > > > > > ### Comment · Reviewer_5E1m · 2023-08-19
> > > > > >
> > > > > > Thank the authors for their detailed responses. I will increase my rating; however, the camera-ready version of this manuscript still requires careful revisions.

---

> > > > > > > ### Author Response · Authors · 2023-08-19
> > > > > > >
> > > > > > > We sincerely thank the reviewer for the constructive suggestions and feedbacks.
> > > > > > >
> > > > > > > Should you have any further discussions on this submission or related topics, we are glad to discuss.
> > > > > > >
> > > > > > > Regards,
> > > > > > >
> > > > > > > -Authors

---

### Author Rebuttal · Authors · 2023-08-10

We thank all reviewers for the valuable feedbacks and constructive comments. We have prepared the response and revised the manuscript accordingly to address your concerns. The major concerns and the corresponding answers/explanations/clarifications are as follows:

1. Related works (5E1m, WXnS, 7vyh) and datasets (WXnS, 7vyh): we add the experiments to compare our method with Yang et al (domain adaptation for MIL, mentioned by 5E1m), HHOT (OT-method for WSI, mentioned by WXnS) and SISH (Nearest neighbor search method, mentioned by WXnS) and will include the missing citations mentioned in the related works. RAM-MIL outperforms the related methods in the experiments.
We also add experimental results on new datasets including TCGA-NSCLC, CPTAC-LSCC and CPTAC-UCEC (WXnS). New experiments with Camelyon17 multi-class tumor stage classification and domain adaptation between hospitals from Camelyon17 (7vyh) are included. Please check the one-page pdf for details.
2. Intrinsic dimension and dimensionality reduction (5E1m, yGH7, Q4PZ and 7vyh): we made additional explanations about intrinsic dimension which is misunderstood as normal dimension. The data are assumed to lie on a low-dimensional manifold with an intrinsic dimensionality, which is lower than the normal dimensionality.
3. Efficiency of retrieval (yGH7, q4PZ, 7vyh) and percentage of patches to calculate OT (yGH7, 7vyh): we test the retrieval efficiency of different variants of RAM-MIL - L2, Hausdorff, approximate-OT, full-OT, which have trade-off in performance and efficiency. Approximate-OT provides a decent performance with a high retrieval efficiency. Users could select the distance metrics based on the need.
During the rebuttal period, we found that using a fixed attention probability to select few patches is technically sound. Using 10% patches might saturate and the performance does not increase.
4. Others: Dimension manipulation with pre-trained model (5E1m), ABMIL with or without linear probe (WXnS), online retrieval (Q4PZ) and dynamic evaluation (7vyh) are answered with new experimental results accordingly. The minor problems like typos will be updated in the manuscript.

We hope that our response can address the mentioned weaknesses and concerns, and the reviewers could raise their scores accordingly.

---

### Author Response · Authors · 2023-08-11
**Look forward to post-rebuttal feedbacks!**

Dear AC and reviewers:

Thanks again for all of your constructive comments, which helped us improve the quality and clarity of the paper!

Our rebuttal has been posted for a while. We have not heard any post-rebuttal response yet. Please do not hesitate to let us know if there are any additional clarifications that we can provide, as we would like to convince you of the merits of the paper.

We appreciate your responses. Thanks!

Sincerely,
Authors

---

### Decision · Program_Chairs · 2023-09-21

**Decision:**

Accept (poster)

**Comment:**

Four out of five reviewers recommend acceptance, while 2 increased their score after rebuttal. This paper augmenta MIL with NN-based retrieval to improve performance of an important application. Although during discussion some of the acceptance reviewers expressed that they think the paper is borderline with novelty weaknesses beyond the medical-imaging domain, they all still recommended acceptance.The ACs do not have strong reason to overturn the reviewers recommendation.

The authors should incorporate the feedback and discussions in the final manuscript.